



# Seasonal dynamics and regional distribution patterns of CO₂ and CH₄ in the north-eastern Baltic Sea

Silvie Lainela[1], Erik Jacobs[2], Stella-Theresa Stoicescu[1], Gregor Rehder[2] and Urmas Lips[1]

[1]Department of Marine Systems, Tallinn University of Technology, Tallinn, 12618, Estonia
[2]Leibniz Institute for Baltic Sea Research Warnemünde, Rostock, 18119, Germany

*Correspondence to*: Silvie Lainela (silvie.lainela@taltech.ee)

**Abstract.** Significant research has been carried out in the last decade to describe the $CO_2$ system dynamics in the Baltic Sea. However, there is a lack of knowledge in this field in the NE Baltic Sea, which is the main focus of the present study. We analysed the physical forcing and hydrographic background in the study year (2018) and tried to elucidate the observed patterns

of surface water $CO_2$ partial pressure ($p$CO₂) and methane concentrations ($c$CH₄). Surface water $p$CO₂ and $c$CH₄ were calculated from continuous measurements during six monitoring cruises onboard R/V Salme, covering the Northern Baltic Proper (NBP), the Gulf of Finland (GoF) and the Gulf of Riga (GoR) and all seasons in 2018. The general seasonal $p$CO₂ pattern showed oversaturation in autumn-winter and undersaturation in spring-summer in all three areas, but it locally reached the saturation level during the cruises in April, May and August in the GoR and in August in the GoF. $c$CH₄ was oversaturated

during the entire study period, and the seasonal course was not well exposed on the background of high variability. Surface water $p$CO₂ and $c$CH₄ distributions showed larger spatial variability in the GoR and GoF than in the NBP for all six cruises. We linked the observed local maxima to river bulges, coastal upwelling events, fronts, and occasions when vertical mixing reached the seabed in shallow areas. Seasonal averaging over the $CO_2$ flux based on our data suggest a weak sink for atmospheric $CO_2$ for all basins, but high variability and the long periods between cruises (temporal gaps in observation)

preclude a clear statement.

## 1 Introduction

Carbon dioxide ($CO_2$) and methane ($CH_4$) are important atmospheric greenhouse gases influencing the global climate. Changes in the levels of these trace gases are monitored in comparison with the pre-industrial era; however, precise and systematic atmospheric $CO_2$ and $CH_4$ measurements were not started before the late 1950s and early 1980s, respectively (Keeling et al.,

2009; Dlugokencky et al., 1994). In the recent decade (2012-2021), the atmospheric $CO_2$ growth rate was $5.2 \pm 0.02$ GtC yr⁻¹ (Friedlingstein et al., 2022). Atmospheric concentration of $CH_4$ remained nearly constant from the late 1990s through 2006, but resumed increasing since then, at an average rate of $7.6 \pm 2.7$ ppb yr⁻¹ estimated for 2010–2019 (Canadell et al., 2023). Methane has large emissions from both natural (e.g. wetlands) and anthropogenic (e.g. enteric fermentation, manure treatment, fossil fuel exploitation) sources, but a clear demarcation of their nature is difficult (Canadell et al., 2023).



There is continuous gas exchange between the atmosphere and the marine environment. The exchange on the air-sea interface is controlled by the air-sea difference in gas concentrations ($CO_2$ or $CH_4$) and by the efficiency of the transfer processes. The solubility of gases in water and, thus, their air-sea exchange also depends on water temperature and salinity. The efficiency of air-sea exchange can be represented by the resistance of the surface and expressed in terms of a transfer velocity (Rutgersson

et al., 2011), which is commonly parameterised as a function of wind speed (e.g. Wanninkhof 2014; Gutiérrez-Loza et al., 2021).

In the Baltic Proper, the seasonal cycle of $CO_2$ is characterised by changing saturation levels between different seasons: oversaturation during autumn and winter and considerable undersaturation during spring and summer (Thomas and Schneider,

1999). Spring and summer periods are characterised by two distinct minima attributed to the spring phytoplankton bloom and the cyanobacteria bloom in midsummer, respectively (Schneider et al., 2014; Schneider and Müller, 2018). Understanding the surface water $CO_2$ dynamics in the Baltic Sea is becoming increasingly important since it is tightly linked to the biogeochemical processes, including primary production and nutrient (nitrogen and phosphorus) dynamics. In addition to the exchange at the air-sea interface and biological processes, the $CO_2$ system of surface waters in the Baltic Sea is influenced by the changes in

hydrographic conditions, e.g. waves, currents, vertical stratification and mixing, upwelling/downwelling, fronts, etc. (e.g. Jacobs et al., 2021).

Methane is formed during the decomposition of organic material by microbial methanogenesis. $CH_4$ generated in the sediments that is not consumed at the sediment/water interface can diffuse into the water column and be transported over large areas of

the Baltic Sea (e.g. Gülzow et al., 2014). $CH_4$ is consumed while approaching the surface water due to methane oxidation at the redoxcline and in the oxygenated water column above (Schmale et al., 2010; Jakobs et al., 2013 & 2014). This leads to strong vertical stratification with elevated concentrations in the sub-redoxcline layer and concentrations near atmospheric equilibrium at the sea surface (e.g. Schmale et al., 2010). Methanogenesis is generally more prevalent in shallower coastal regions due to the higher organic matter content (Valentine, 2002). In coastal areas, the controlling factors for the seasonal

variations of methane emission are the sediment organic matter content (Heyer and Berger, 2000) and temperature (Borges et al., 2018). In areas where the water column is relatively shallow and constantly mixed, $CH_4$ may escape into the atmosphere more readily. In general, the Baltic Sea is a source of atmospheric $CH_4$ (Bange et al., 1994; Gülzow et al., 2013), with the majority of methane emissions coming from shallow coastal areas (e.g. Roth et al., 2022). Outgassing can be intensified as a consequence of high water temperatures (Humborg et al., 2019) and processes driving vertical transport and mixing, e.g.

upwelling events (Jacobs et al., 2021). Production in the upper, oxygenated water column might also contribute to or even govern methane sea-air fluxes (Schmale et al., 2018; Stawiarski et al., 2019).



This work is the first extensive trace gas ($CH_4$ and $CO_2$) study in the north-eastern Baltic Sea area, with the main focus on the southern Gulf of Finland (GoF) and the Gulf of Riga (GoR), allowing the assessment of surface layer trace gas and carbon

system dynamics in the region. The main aim of our work is to describe the spatial variability and seasonal dynamics of $CO_2$ and $CH_4$ and compare these patterns with the better studied Northern Baltic Proper (NBP) (Schneider et al., 2014; Schneider and Müller, 2018; Jakobs et al., 2014; Gülzow et al., 2013). We analysed the physical forcing and hydrographic and biological background in the study year (2018) and made an effort to link the observed patterns of $CO_2$ and $CH_4$ to these drivers.

The questions we try to answer are: Is the seasonal cycle of $CO_2$ and $CH_4$ in the southern GoF and GoR similar to that in the NBP? Can we elucidate regional differences in $CO_2$ and $CH_4$ dynamics due to river discharges, water depth and mixing, fronts and upwelling events, or other hydrographic features? Do the regional variations in $CO_2$ and $CH_4$ dynamics result in differences in yearly fluxes of these gases between the sub-basins? Our analysis is based on measurements during six reoccurring cruises of the Estonian monitoring programme in the north-eastern Baltic Sea (Fig. 1).

**2 Study area**

The Baltic Sea is a brackish, semi-enclosed sea in northern Europe. High freshwater runoff from the catchment area and sporadic saline water inflows from the North Sea maintain horizontal gradients and vertical stratification (e.g. Leppäranta and Myrberg, 2009). A quasi-permanent halocline exists at depths of 60-70 m in the deeper basins, and a seasonal thermocline develops at depths of 10-20 m from spring to autumn. The present study covers the following Baltic Sea sub-basins (Fig. 1):

the NBP (we assign to it also a small fraction of the Eastern Gotland Basin), the GoF and the GoR.

The Northern Baltic Proper is the deepest sub-basin with a maximum depth of about 200 m and very variable topography and coastline. The quasi-permanent halocline separates oxygenated waters in the upper layers and hypoxic/anoxic waters below the halocline. However, no quasi-stationary horizontal gradients of environmental parameters exist in the surface layer. The

general circulation pattern in the surface layer is considered mostly cyclonic (e.g. Placke et al., 2018). However, the presence of the northward boundary current along the eastern coasts depends on local wind patterns and, as a consequence, downwelling and (less often) upwelling events and associated mesoscale currents may occur (Liblik et al., 2022).

The Gulf of Finland is an elongated basin (length about 400 km, width varies between 48 and 135 km) with a mean depth of

37 m (depths >100 m in the western gulf). Due to the direct connection to the NBP in the west and the largest freshwater discharge in the eastern end (Neva River), surface layer salinity decreases from about 6-7 g kg$^{-1}$ at the entrance area to <2 g kg$^{-1}$ in the easternmost area (e.g. Alenius et al., 1998). During winter, the water body is mixed fully in the shallower areas and down to the depth of the quasi-permanent halocline in deeper areas (Alenius et al., 2003). Hypoxic conditions are often observed below the halocline in the deeper areas (e.g. Stoicescu et al., 2019). General circulation in the surface layer is



classically considered to be cyclonic (Andrejev et al., 2004), but could be seasonally variable (Maljutenko and Raudsepp, 2019). Energetic mesoscale features – eddies, fronts, upwelling events, etc. (Pavelson, 2005; Lips et al., 2016a; Kikas and Lips, 2016) may frequently occur. The largest freshwater source along the research vessel (R/V) track analysed in the present study is the Narva River in the south-eastern GoF. Depending on the seasonally varying runoff and local wind conditions, the river water spreads towards the open sea or along the coast and mixes with the gulf water masses (Laanearu and Lips, 2003).


The Gulf of Riga is a semi-enclosed shallow basin with a mean depth of 26 m (Ojaveer, 1995) and a maximum depth of the central basin of 56 m (Stiebrins and Väling, 1996). Freshwater discharge originates mostly from five larger rivers (Daugava, Lielupe, Gauja, Pärnu, and Salaca) in the southern and eastern parts of the gulf (Yurkovskis et al., 1993). Saltier waters from the Baltic Proper enter the gulf via the Irbe Strait in the west (about 70-80% of water exchange) and the Suur Strait in the north

(Astok et al., 1999). The whole-basin circulation in the surface layer depends on the prevailing wind pattern – it is mostly cyclonic, but anti-cyclonic in summer (Lips et al., 2016b). Anti-cyclonic circulation could also prevail in the southern gulf, connected to the discharges from the Daugava and Lielupe rivers (Soosaar et al., 2016). Due to the shallowness of the basin, the water column is fully mixed in autumn-winter. Thermal stratification starts to develop in April and decays in October-December, depending on the water depth and yearly variable meteorological conditions (Skudra and Lips, 2017; Stoicescu et

al., 2022). Near-bottom seasonal hypoxia can be observed in the deeper areas of the central gulf (Stoicescu et al., 2022).

The research vessel track reached the southern gulf close to the largest river discharges (Daugava and Lielupe) and the Pärnu Bay, which is shallow and under the influence of the Pärnu River discharge. The measurements were also conducted in the Väinameri Sea – the shallow and sheltered sea area (average depth of 5-10 m) between the mainland and the western Estonian

islands.





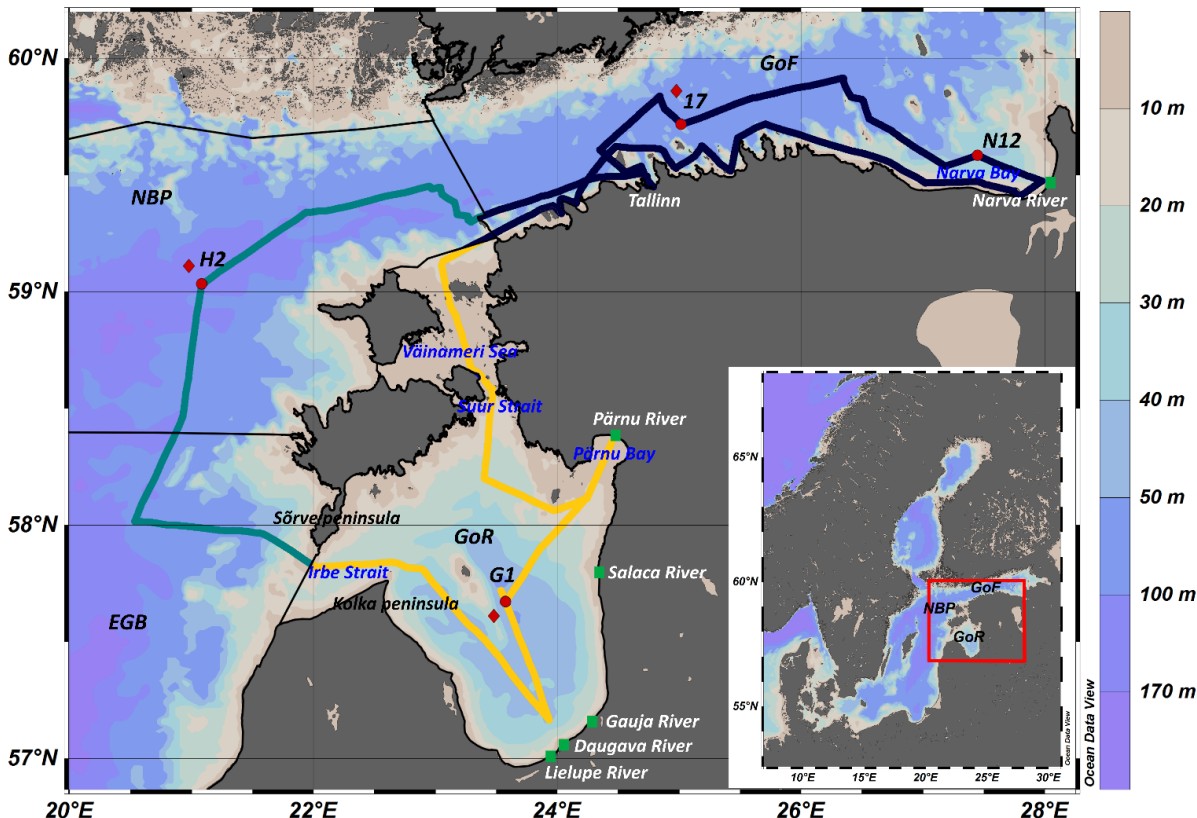

**Figure 1: Map of the study area with bottom topography: GoF – Gulf of Finland (blue cruise track), NBP – Northern Baltic Proper**
**(green) and GoR – Gulf of Riga (yellow) (according to HELCOM sub-basins division, marked with black line). Green-filled squares**
**denote river runoffs. Red-filled circles represent the locations of the most characteristic stations of the sub-basins. Red-filled**
**diamonds denote the closest meteorological ERA5 data grid points to the sub-basins' most characteristic stations. This map was**
**generated using Ocean Data View 5.6.3 software (Schlitzer, 2022).**

## 3 Material and methods

The spatial variability and seasonal dynamics of $CO_2$ and $CH_4$ in three sub-basins of the north-eastern Baltic Sea in the study
year are characterised. The results are analysed considering the background meteorological and hydrographic conditions, e.g.,
upper mixed layer (UML) temperature and depth, bottom depth vs UML depth, fronts and upwelling events, and seasonal and
spatial patterns of Chl $a$ distribution in the surface layer. Also, $CO_2$ and $CH_4$ fluxes are estimated for all studied areas.
Measurement approaches, additional data sources and calculation methods are described below.

### 3.1 Meteorological information


Meteorological conditions were evaluated by ERA5 comprehensive reanalysis data (from Copernicus Climate Data Store;
Hersbach et al., 2019). Surface net solar radiation, air temperature at 2 m above the sea surface and winds at 10 m above the
sea surface were extracted for the positions closest to the monitoring stations 17, G1 and H2 in the GoF, GoR and NBP,



respectively (see Fig. 1). For the comparison of the study year (2018) and the long-term (1979-2018) meteorological
conditions, ERA5 data at station NBP were used. Monthly average reanalysis values in 2018 are presented against the long-term monthly averages and variability, characterised by standard deviations and minimum and maximum values. Average wind vectors and wind roses were calculated for the selected stations in the GoF, GoR and NBP (Fig. 1) for the cruise periods using 2018 hourly reanalysis data. The presented wind characteristics during the cruises represent seven-day periods ending at the cruise termination date, i.e. periods containing 1-2 days before the cruise up to its end.

**3.2 Continuous surface water measurements aboard R/V Salme**

The measurements were conducted using a flow-through system (Ferrybox) onboard R/V Salme during six monitoring cruises in 2018: on 8-12 January, 16-20 April, 28 May-2 June, 9-13 July, 22-27 August, and 22-28 October. The Ferrybox by Go-systemelektronik was equipped with an SBE38 sensor for temperature, an SBE45 MicroTSG sensor for temperature and conductivity, WetLabs *ECO* FL and Turner Design Cyclops-7 sensors for chlorophyll *a* fluorescence and a digital optode by
PONSEL for dissolved oxygen measurements. The Ferrybox water intake was located at a depth of 2 m. The sampling interval was 1 minute, corresponding to a nominal spatial resolution of about 250 m while the vessel was moving with its normal cruising speed of 8-9 knots.

The Ferrybox was supplemented with the equipment for trace gas ($CO_2$ and $CH_4$) measurements using an equilibrator setup.
During the first cruise in January, a LI-COR 6262 $CO_2$/$H_2O$ instrument coupled to the headspace of a glass equilibrator (similar to Gülzow et al., 2011) was used. During the other cruises, the setup was similar to the MESS presented in Sabbaghzadeh et al. (2021) using a Los Gatos Research $CH_4$/$CO_2$ analyser, but with a lower water flow of around 2.5–4.5 L min$^{-1}$. In July and October, the e-folding response times of the setup were exemplarily determined to be 720 s and 790 s for $CH_4$ and 35 s and 52 s for $CO_2$; the lower values in July illustrate the influence of higher water temperature (ca. 18 vs. 12.5 °C) and higher water
flow (ca. 3.6 vs. 3.1 L min$^{-1}$). Apart from January, an additional Microx 4 oxygen meter PSt7 optode measured dissolved oxygen in the water supply line of the equilibrator.

Atmospheric pressure was measured during the entire cruise, and measurements of atmospheric $CO_2$ and $CH_4$ (as mole fractions in ppm/ppb) were performed 1-2 times per cruise. For this, the gas supply of the trace gas analyser was switched
from the equilibrator to a long tubing used to sample air on the windward side of the upper deck.

The data series contain surface layer temperature, salinity, chlorophyll *a* concentration (Chl *a*), partial pressure of $CO_2$ ($pCO_2$), dissolved oxygen concentration and concentration of $CH_4$ ($cCH_4$) with a spatial resolution of about 250 m along the cruise tracks with a length of about 1500 km each, covering three Baltic Sea sub-basins.



### 3.3 Quality assurance and processing of continuous flow-through data

A two-step calibration procedure was followed for the Ferrybox Chl $a$ fluorescence data. First, a linear regression was found between Chl $a$ fluorescence data from the CTD (Ocean Seven 320$plus$, Idronaut s.r.l., with Seapoint fluorometer) and Chl $a$ concentrations determined in the laboratory from the water samples collected at respective stations and depths. Chl $a$ concentration in the laboratory was determined optically by spectrophotometry (HELCOM, 2017). Afterwards, a linear regression between the calibrated Chl $a$ data from CTD at 2 m depth and Ferrybox Chl $a$ fluorescence data was found. This calibration procedure was performed separately for each cruise.

The same two-step calibration procedure was used for dissolved oxygen measurements during the cruises in January and April. Dissolved oxygen concentrations from water samples were determined electrochemically using a dissolved oxygen meter (OX 400 1 DO analyzer; WWR International, LCC), also taking into account a salinity correction. For other cruises, the Microx 4 oxygen meter PSt7 optode data was used. Oxygen partial pressures ($pO_2$) from the PSt7 optode were post-calibrated using discrete sample measurements conducted at 1 m depth at monitoring stations. The procedure was followed separately for each cruise, assuming that the first and last discrete measurements were representative for the start and end of each cruise.

Measured $CO_2$ and $CH_4$ mole fractions ($xCO_2$/$xCH_4$) were post-calibrated using a near-atmospheric standard gas (398.49 ppm $CO_2$, 1.91 ppm $CH_4$, matrix: ambient air). These target measurements were performed at the beginning and end of each cruise and almost every day at sea to achieve a drift correction if necessary. Measured $xCO_2$ and $xCH_4$ were converted into dry-air values based on water mole fractions measured by the same instrument. From these, the partial pressures ($pCO_2$/$pCH_4$) were calculated assuming 100 % humidity in the equilibrator headspace (water vapour pressure by Weiss and Price, 1980). $pCO_2$ was temperature-corrected to account for water warming from the inlet to the equilibrator (Takahashi et al., 1993). $CH_4$ partial pressure data were converted to concentration ($cCH_4$) using the solubility constants given in Wiesenburg and Guinasso (1979). All equilibrator data were averaged using a 1-minute rolling mean to match the temporal resolution of other Ferrybox parameters.

Despite the fact that we actually recorded mole fractions ($xCO_2$ and $xCH_4$), we report our $CO_2$ data as $pCO_2$ and $CH_4$ data as $cCH_4$, considering that these units are usually reported in studies addressing the respective gases. Accordingly, the atmospheric data were displayed as atmospheric partial pressure for $CO_2$ or saturation concentration calculated from temperature and salinity for $CH_4$.

Surface flow-through $pCO_2$ and $cCH_4$ data recorded at the monitoring stations were excluded, using a speed of the vessel of less than 0.6 knots as criterion. This was necessary because during profiling and water sampling, the sampling device or ship propulsion could bring up sub-surface water, which caused artificial spikes in $pCO_2$ and $cCH_4$ signals.



**3.4 CTD profiles and upper mixed layer depth**

Vertical profiles of temperature, salinity, Chl *a* fluorescence and dissolved oxygen were recorded at the monitoring stations
using the CTD probe (Ocean Seven 320plus; Idronaut s.r.l). Salinity and density anomaly are shown as absolute salinity (g kg$^{-1}$) and sigma-0 (kg m$^{-3}$), respectively, and were calculated using the TEOS-10 equation of state (IOC et al., 2010). The depth
of the UML was determined from the CTD profiles at the monitoring stations for all six monitoring cruises. The UML depth
was defined according to Liblik and Lips (2012) as the minimum depth, where $\rho_z - \rho_3 > 0.25$ kg m$^{-3}$, where $\rho_z$ is the density
anomaly at depth z and $\rho_3$ at depth 3 meters.

**3.5 Air-sea $CO_2$ and $CH_4$ flux calculations**

Air-sea gas exchange calculations were performed using the FluxEngine toolbox. The FluxEngine toolbox is an open-source
software package described in more detail by Shutler et al. (2016) and Holding et al. (2019).

The $CO_2$ fluxes were calculated using a rapid model approach (Woolf et al., 2016):
$$F = k(\alpha_W pCO_{2W} - \alpha_A pCO_{2A}),\tag{1}$$
where *F* (g C m$^{-2}$ day$^{-1}$) denotes the flux across the interface, *k* the gas transfer velocity, *α* the solubility of gas in the
seawater/air-sea interface (subscripts *W* and *A*, accordingly) and *p*$CO_2$ partial pressure of $CO_2$ in the sea surface
water/atmosphere (subscripts *W* and *A*, accordingly).

Methane fluxes were calculated using the same approach as for $CO_2$ fluxes:
$$F = k(\alpha_W cCH_{4W} - \alpha_A cCH_{4A}),\tag{2}$$
where *c*$CH_4$ is concentration of $CH_4$ in the surface sea water/atmosphere (subscripts *W* and *A*, accordingly).

In order to accurately describe the fluxes and the carbon budget, it is essential to include relevant processes to the air–sea $CO_2$
and $CH_4$ flux parametrisation. The sensitivity analysis of the gas transfer velocity in the Baltic Sea (Gutiérrez-Loza et al.,
2021) used different parametrisations of the gas transfer velocity to evaluate the effect of other relevant processes in addition
to wind speed on the net $CO_2$ flux at regional and sub-regional scale. In the Estonian sea area, negligible differences in the
average net $CO_2$ flux were observed when using the different gas transfer parametrisations relative to the wind-based
parametrisation (Nightingale et al. 2000). Therefore Nightingale et al. (2000) was used for the gas transfer velocity
parametrisation for both $CO_2$ and $CH_4$ in our study:
$$k = (0.222U_{10}^2 + 0.333U_{10})\sqrt{600/Sc},\tag{3}$$
where $U_{10}$ is the wind speed at 10 m above the sea surface and *Sc* is the Schmidt number. Schmidt number parametrisation was
based on Wanninkhof (2014).





## 4 Results

### 4.1 Meteorological conditions

Meteorological conditions in the Baltic Sea area in 2018 were characterised by warmer than long-term average air and sea surface temperatures (Hoy et al., 2020; Humborg et al., 2019). Net solar radiation was above the average seasonal curve from February to September, with the maximum positive deviation in May (Fig. 2). In accordance with the latter, the monthly mean air temperature exceeded the long-term average from April until the end of the year. Except for June, the monthly mean wind speed in the spring and summer of 2018 was lower than the long-term average. All these meteorological parameters predict that the sea surface temperature should have been higher and seasonal vertical stratification stronger than on average due to the increased positive buoyancy flux and weak wind-induced mixing.

The winds from west and south-west prevailed during and before the cruises in January, April and August (Fig. 2), which is in accordance with the general airflow in the study area. During the cruises in July and October, the wind direction was generally from north or north-east, while weak winds from the same direction prevailed in May-June. Note that the northerly and north-easterly winds in July and October were favourable for the upwelling development along the southwestern coast of the Gulf of Finland and the eastern coasts in the Northern Baltic Proper.

The variability in wind speed and direction between the three basins and within the cruise periods is presented by the wind roses (see Fig. 3, where wind roses for three cruise periods with larger wind forcing and spatial variability are presented). The winds were mostly from one direction during the cruise in August, and a wider spread was characteristic for the cruise periods in July and October. In July, mostly two directions prevailed – easterly winds, which could cause upwelling events along the entire southern coast of the Gulf of Finland, and north-westerly winds, which are upwelling favourable for the eastern coasts of the NBP. In October, the spread of directions was the largest, but the strongest winds with speeds exceeding 15 m s$^{-1}$ in the GoF and NBP were from the north-east. In the GoR, wind speeds were generally lower than in the other two basins.



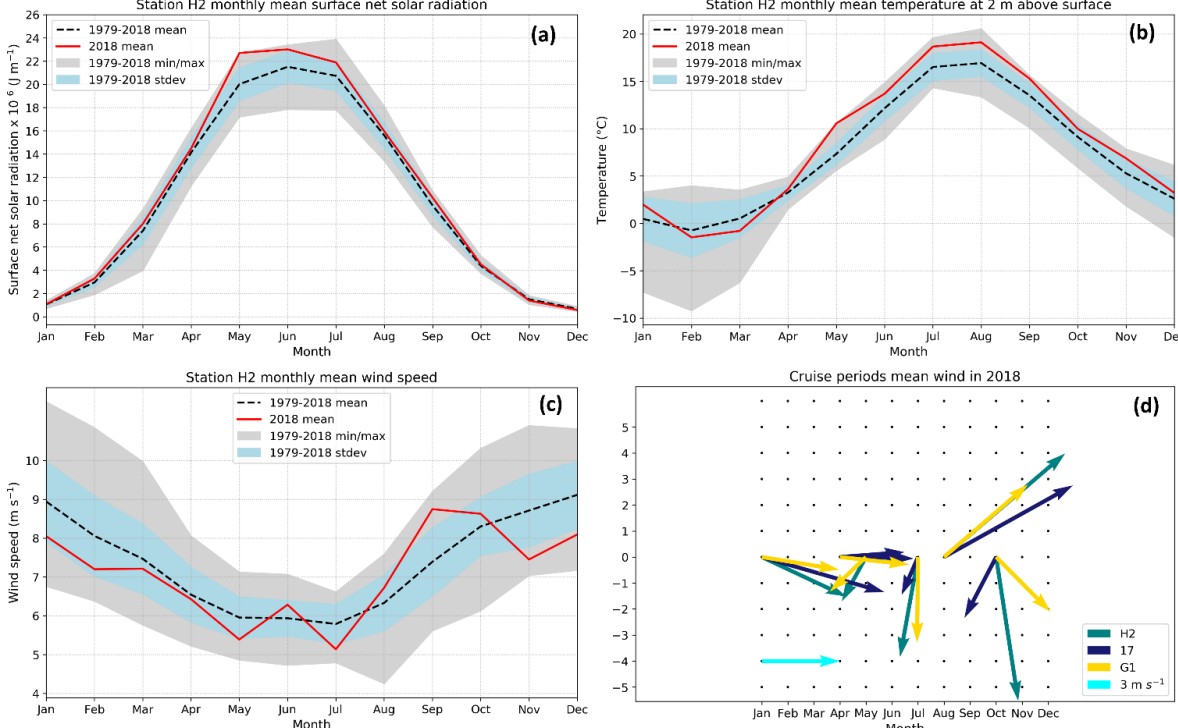

**Figure 2: Monthly average (a) surface net solar radiation, (b) air temperature and (c) wind speed in 2018 (red line) compared with the long-term averages (dashed black line), standard deviations (blue area) and minimum-maximum values (grey area) in the NBP (station H2) for the period 1979-2018. (d) Cruise-period average wind vectors in the NBP (H2, green), GoF (17, blue) and GoR (G1, yellow). See the locations of stations and model grid points for data extraction in Fig. 1.**





**Figure 3: Cruise-period wind roses in May-June (left column), July (middle column) and August (right column) 2018 in the NBP (upper row), GoF (middle row) and GoR (lower row) based on the hourly ERA5 data extracted from a grid point close to the monitoring stations H2, 17 and G1, respectively (see locations in Fig. 1). Radial axis maximum is 30 (out of 168).**

**4.2 Spatial variability**

Surface water $p$CO$_2$ and $c$CH$_4$ distributions along the R/V track show larger spatial variability in the GoR and GoF than in the NBP for all six cruises (Figs. 4-9). Although the general seasonal course with CO$_2$ oversaturation in the surface layer in winter and autumn and undersaturation in spring-summer is evident, $p$CO$_2$ locally exceeded the atmospheric equilibrium level also during the cruises in April, May and August. The latter is mostly valid for the GoR but also for the GoF in August. $c$CH$_4$ was oversaturated in the surface layer during the whole study period, with prominent local peaks of $c$CH$_4$ in the GoR and GoF.



In January (Fig. 4), surface water $pCO_2$ along the cruise track (Fig. 4c) did not show remarkable regional differences. Values fluctuated within the range of 425 – 550 µatm and were oversaturated in all monitored areas. Almost in all areas, the water column was well-mixed down to the seabed or permanent halocline. Note that in contrast to the other cruises, the vessel did not visit mouth areas of the rivers (neither in the GoR nor in the GoF) and $cCH_4$ was not measured in January.

The cruise in April (Fig. 5) mapped $pCO_2$ and $cCH_4$ distributions in the period of the onset of seasonal stratification and spring bloom in different development phases. The surface water $pCO_2$ (Fig. 5c) was mostly below the atmospheric partial pressure but reached equilibrium with the atmosphere in the western and central GoR. Low Chl $a$ and oxygen concentrations indicate that the spring bloom was yet in its initial phase in this area (Fig. 5d). Also, the water column was mixed almost down to the seabed in this sea area. The $pCO_2$ values were clearly lower in the eastern part of the open GoR and the shallow Pärnu Bay and

the Väinameri Sea. These $pCO_2$ minima (down to 60 µatm) were associated with increased temperature and the highest Chl $a$ and oxygen concentrations in these areas, while the influence of the Pärnu River was visible via a local peak in the $pCO_2$ (up to 397 µatm).

    From the western GoF towards the central GoF, a Chl $a$ increase from <9 to 15 mg m$^{-3}$ was accompanied by slightly higher

oxygen and lower $pCO_2$ values (about 130 µatm). Interestingly, high Chl $a$ concentrations up to 16 mg m$^{-3}$ were mapped along the south-eastern coast of the GoF, but $pCO_2$ values in this area remained on a higher level, around 250 µatm, than in other regions with a similar surface layer Chl $a$ content. Note that the water column along the coast, except close to the Narva River mouth, was well mixed down to the seabed. Elevated $cCH_4$ were measured, similar to the $pCO_2$ distribution, in the western and central GoR (27-37 nmol L$^{-1}$) and in the Pärnu Bay close to the mouth of the Pärnu River (40 nmol L$^{-1}$). The high methane

concentration in the central GoR could be related to the absence of vertical stratification and the almost fully mixed water column. Elevated $cCH_4$ were also measured along the south-western coast of GoF (26 nmol L$^{-1}$; marked as SW GoF in Fig. 5e), where the water column was mixed down to the seabed, and close to the mouth of the Narva River (up to 30 nmol L$^{-1}$). The direct influence of the Pärnu River and Narva River was expressed by the $cCH_4$ peaks.

The cruise at the end of May and beginning of June (Fig. 6) coincided with the phytoplankton summer minimum, while the water column was characterised by unusually high sea surface temperatures (13-18 °C; Fig. 6b) and shallow upper mixed layer (Fig 6a, on average <6 m in the GoR and 8 m in the central GoF). The $pCO_2$ values had decreased to the minimum along most of the ship track, varying mostly between 50 and 100 µatm, while moderately higher (reaching 200 µatm) than background $pCO_2$ values were registered occasionally in the NBP accompanied by locally lower temperatures at 2 m depth and in the NBP-

GoR transition area, the Irbe Strait. $CO_2$ oversaturation was locally recorded in the GoR shallowest areas – the Pärnu Bay and the Väinameri Sea. In the Narva Bay, no distinct Narva River impact was registered. Local $cCH_4$ maxima were observed in the shallow bays of the southern and south-western GoF (>80 nmol L$^{-1}$) and in the Irbe Strait, close to the Kolka peninsula





($>35$ nmol L$^{-1}$). Only slightly higher than background $c$CH$_4$ values were measured in the southern GoR close to the mouths of the largest rivers – Daugava and Lielupe. As indicated by the salinity distribution, the vessel almost did not pass the waters

with strong river influence. In contrast, an extensive peak in $c$CH$_4$ was registered in the shallow Pärnu Bay close to the Pärnu River (maximum measured concentration was 232 nmol L$^{-1}$). Local maxima in the shallow Väinameri Sea increased to notable peaks along the south-western coast of GoF. Local maxima of 33 nmol L$^{-1}$ in the Narva Bay were likely due to the Narva River influence.

In mid-July (Fig. 7), the surface waters were undersaturated in CO$_2$ along the entire ship track (Fig. 7c; note that the vessel did not visit the mouth areas of the Pärnu and Narva rivers). The UML has slightly deepened in the GoR (8 m), but was the shallowest in the offshore GoF ($<7$ m). Higher than 5 mg m$^{-3}$ Chl $a$ concentrations were observed in the offshore areas of NBP, north-eastern GoR and central GoF, probably due to the development of the summer bloom. Elevated $p$CO$_2$ values were recorded in parts of the NBP offshore areas (330 µatm) and the coastal sea area, in the Irbe Strait (310 µatm) and the Väinameri

Sea (355 µatm). Local maxima of $c$CH$_4$ up to 43 nmol L$^{-1}$ were observed in the shallow bays of the southern and south-western GoF, including the transition area into the GoF. In comparison with the cruise at the end of May, in July a relatively low $c$CH$_4$ peak (14 nmol L$^{-1}$) was observed in the Irbe Strait, and the influence of large rivers in the southern GoR was almost not detectable.

In August (Fig. 8), CO$_2$ varied around the saturation level. CO$_2$ was undersaturated in most areas of the GoF except the Narva Bay, where the water column was well-mixed down to the seabed, oversaturated in the Väinameri Sea and Pärnu Bay, and undersaturated in the NBP (Fig. 8c). These higher $p$CO$_2$ values were characteristic for the shallow areas and could be only partly related to the river discharge (as in the Narva Bay and the Pärnu Bay). A distinct local maximum in $p$CO$_2$ of 460 µatm was related to the salinity front in the Irbe Strait, as also observed earlier. Local $c$CH$_4$ maxima were observed in the shallow

bays of the southern and south-western GoF and along the south-eastern coast in the Narva Bay. Locally, well-pronounced $c$CH$_4$ peaks with a maximum concentration of 177 nmol L$^{-1}$ were also observed in the Väinameri Sea and Pärnu Bay (probably influenced by the Pärnu River discharge). Increase in $c$CH$_4$ in the Irbe Strait (38 nmol L$^{-1}$) was comparable with the $c$CH$_4$ peak at the end of May cruise.

In October (Fig. 9), surface waters were oversaturated in CO$_2$ almost along the entire ship track (Fig. 9c). Like during the August cruise, $p$CO$_2$ values were lower in the NBP (varying between 400 and 480 µatm) than in the GoF and GoR (varying up to 600 µatm). In contrast to the summer cruises, higher $p$CO$_2$ values were characteristic for the offshore areas in the GoR, and lower values in the shallow coastal sea areas – the Pärnu Bay and the Väinameri Sea. $p$CO$_2$ values slightly exceeding 1200 µatm were registered in connection to an upwelling event in the SW GoF. This upwelling event was caused by the strong

north-westerly winds before and during the cruise (Figs. 2 and 3) and is better seen in salinity than temperature distribution (Fig. 9b). Peaks in $c$CH$_4$ up to 80 nmol L$^{-1}$ were registered in the shallow bays along the southern coast of GoF and also in the



shallow coastal area in the NBP (off Saaremaa Island) that was not visited during the earlier cruises. In the Irbe Strait, $c$CH$_4$ increased up to 15 nmol L$^{-1}$ in October. A clear $c$CH$_4$ peak of 62 nmol L$^{-1}$ was detected in the Pärnu Bay, probably influenced by the Pärnu River discharge. Local maxima in the Väinameri Sea increased to an extensive broad peak (69 nmol L$^{-1}$) in the

upwelling waters in the SW GoF.











**Figure 4: January monitoring cruise (8-12 January): The trajectory is shown on the map and on panel (a) UML depth (light blue bars) and the water column extent below the UML (dark blue bars); vertical grey dashed lines indicate the locations of monitoring stations, the locations of the most characteristic stations of the sub-basins are denoted with red dots. (b) Spatial variability of temperature (left y-axis) and salinity (right y-axis), (c) $CO_2$ partial pressure (left y-axis) and relative saturation (right y-axis), and (d) Chl *a* (left y-axis), dissolved oxygen concentration (left y-axis) and saturation (right y-axis). The x-axis denotes the distance (in km) from the start of the monitoring cruise (Tallinn).**






**Figure 5: April monitoring cruise (16-20 April): The trajectory is shown on the map and on panel (a) UML depth (light blue bars) and the water column extent below the UML (dark blue bars); vertical grey dashed lines indicate the locations of monitoring stations, the locations of the most characteristic stations of the sub-basins are denoted with red dots. (b) Spatial variability of temperature (left y-axis) and salinity (right y-axis), (c) $CO_2$ partial pressure (left y-axis) and relative saturation (right y-axis), (d) Chl *a* (left y-axis), dissolved oxygen concentration (left y-axis) and saturation (right y-axis), and (e) $CH_4$ concentration (left y-axis) and relative saturation (right y-axis). The x-axis denotes the distance (in km) from the start of the monitoring cruise (Tallinn).**







**Figure 6: End of May monitoring cruise (28 May-2 June): The trajectory is shown on the map and on panel (a) UML depth (light blue bars) and the water column extent below the UML (dark blue bars); vertical grey dashed lines indicate the locations of monitoring stations, the locations of the most characteristic stations of the sub-basins are denoted with red dots. (b) Spatial variability of temperature (left y-axis) and salinity (right y-axis), (c) CO₂ partial pressure (left y-axis) and relative saturation (right y-axis), (d) Chl *a* (left y-axis), dissolved oxygen concentration (left y-axis) and saturation (right y-axis), and (e) CH₄ concentration (left y-axis) and relative saturation (right y-axis). The x-axis denotes the distance (in km) from the start of the monitoring cruise (Tallinn). In May, *c*CH₄ signal in the river estuaries was extreme in comparison with the rest of the data, and this signal was cut on the panel to 100 nmol L⁻¹ to properly display the structure within low-concentration areas. Note different scale for temperature in comparison with January and April.**








**Figure 7: July monitoring cruise (9-13 July): The trajectory is shown on the map and on panel (a) UML depth (light blue bars) and the water column extent below the UML (dark blue bars); vertical grey dashed lines indicate the locations of monitoring stations, the locations of the most characteristic stations of the sub-basins are denoted with red dots. (b) Spatial variability of temperature (left y-axis) and salinity (right y-axis), (c) CO₂ partial pressure (left y-axis) and relative saturation (right y-axis), (d) Chl *a* (left y-axis), dissolved oxygen concentration (left y-axis) and saturation (right y-axis), and (e) CH₄ concentration (left y-axis) and relative saturation (right y-axis). The x-axis denotes the distance (in km) from the start of the monitoring cruise (Tallinn).**










**Figure 8: August monitoring cruise (22-27 August): The trajectory is shown on the map and on panel (a) UML depth (light blue bars) and the water column extent below the UML (dark blue bars); vertical grey dashed lines indicate the locations of monitoring stations, the locations of the most characteristic stations of the sub-basins are denoted with red dots. (b) Spatial variability of temperature (left y-axis) and salinity (right y-axis), (c) CO$_2$ partial pressure (left y-axis) and relative saturation (right y-axis), (d) Chl *a* (left y-axis), dissolved oxygen concentration (left y-axis) and saturation (right y-axis), and (e) CH$_4$ concentration (left y-axis) and relative saturation (right y-axis). The x-axis denotes the distance (in km) from the start of the monitoring cruise (Tallinn). In August, *c*CH$_4$ signals in the river estuaries and coastal areas were extreme in comparison with the rest of the data, and these signals were cut on the panel to 100 nmol L$^{-1}$ to properly display the structure within low-concentration areas.**









**Figure 9: October monitoring cruise (22-28 October):  The trajectory is shown on the map and on panel (a) UML depth (light blue bars) and the water column extent below the UML (dark blue bars); vertical grey dashed lines indicate the locations of monitoring stations, the locations of the most characteristic stations of the sub-basins are denoted with red dots. (b) Spatial variability of temperature (left y-axis) and salinity (right y-axis), (c) CO$_2$ partial pressure (left y-axis) and relative saturation (right y-axis), (d) Chl *a* (left y-axis), dissolved oxygen concentration (left y-axis) and saturation (right y-axis), and (e) CH$_4$ concentration (left y-axis) and relative saturation (right y-axis). The x-axis denotes the distance (in km) from the start of the monitoring cruise (Tallinn). In October, *c*CH$_4$ signal in the coastal areas was extreme in comparison with the rest of the data, and this signal was cut on the panel to 100 nmol L$^{-1}$ to properly display the structure within low-concentration areas. Note different scale for CO$_2$ partial pressure and relative saturation in comparison with the rest of the cruises.**

## 4.3 Seasonal variability

Seasonal variability of *p*CO$_2$ in 2018 (Fig. 10d) follows the general seasonal course in all analysed sub-basins with high values in winter, a decrease in spring, a minimum in late spring, and an increase in autumn (Table 1). The *p*CO$_2$ decrease in spring coincidences with the highest Chl *a* and dissolved oxygen concentrations in April. Based on decreased Chl *a* concentrations from mid-April to the end of May (Fig. 10e; Table 1), the early-summer minimum of phytoplankton biomass was evident. It is also at the end of May – early June when the *p*CO$_2$ seasonal minimum in all sub-basins appeared (Table 1). During midsummer, relatively high Chl *a* concentrations were recorded (Fig. 10e; Table 1) and dissolved oxygen concentrations stayed moderately oversaturated. *p*CO$_2$ values increased from July on, reaching oversaturation almost everywhere along the cruise track by the October cruise. No second *p*CO$_2$ minimum during summer nor a relative maximum between the two usually expected minima in spring and late summer were detected.

For the evaluation of the seasonal course of surface water methane concentrations, *c*CH$_4$ median values were analysed (Fig. 10h and Table 1). In all three sub-basins, the highest median concentrations of 13.7 nmol L$^{-1}$ in the GoR, 11.5 nmol L$^{-1}$ in the GoF and 7.6 nmol L$^{-1}$ in the NBP were determined in April (note we do not have winter data), after which the concentrations started to decrease. The minimum level was reached in the GoF and GoR in July (median concentrations were 7.9 nmol L$^{-1}$ and 4.5 nmol L$^{-1}$, respectively) and in the NBP in August (3.9 nmol L$^{-1}$). It was followed by an increase in concentrations by October, but the values did not reach yet the levels observed in April.

Although high *c*CH$_4$ values represent only a small part of acquired data, it is worthwhile to mark some seasonal changes in variability. The highest *c*CH$_4$ variations were observed in May, August and October in the GoF and in April, May and August in the GoR. It shows that although the average seasonal course in the GoF and GoR was similar to the NBP, regions existed in the GoF and GoR with locally high methane concentrations (extremes exceeding 100 nmol L$^{-1}$).





**Figure 10: Median and 5/95 percentile values of (a) UML depth, (b) temperature, (c) salinity, (d) CO₂ partial pressure, (e) Chl _a_, (f) dissolved oxygen and (g) saturation, (h) CH₄ concentration and (i) relative saturation. Whiskers denote min and max values. In May, August and October cruises, _c_CH₄ signals in the river estuaries and coastal areas were extreme in comparison with the rest of the**



data, and these signals are not seen on the plot (y-axis maximum is 100 nmol L$^{-1}$ to properly display the pattern in low-concentration areas).



**Table 1.** Median / mean values of UML depth, temperature (T), salinity (S), CO$_2$ partial pressure ($p$CO$_2$), Chl $a$, dissolved oxygen (O$_2$) and saturation (O$_2$ sat.), CH$_4$ concentration ($c$CH$_4$) and relative saturation (CH$_4$ rel.) for the Estonian sea area sub-basins in 2018.

| | | UML (m) | T (°C) | S (g kg$^{-1}$) | $p$CO$_2$ (µatm) | Chl $a$ (mg m$^{-3}$) | O$_2$ (mg l$^{-1}$) | O$_2$ sat. (%) | $c$CH$_4$ (nmol L$^{-1}$) | CH$_4$ rel. sat. (%) |
|---|---|---|---|---|---|---|---|---|---|---|
| **GoF** | Jan | 35 / 38 | 3.5 / 3.5 | 5.8 / 5.9 | 511 / 511 | 1.0 / 1.0 | 11.6 / 11.7 | 92 / 92 | – | – |
| | Apr | 15 / 15 | 2.2 / 2.2 | 5.1 / 5.1 | 141 / 178 | 9.0 / 9.7 | 15.0 / 14.9 | 114 / 112 | 11.5 / 13.8 | 2.5 / 3.0 |
| | May | 8 / 7 | 14.0 / 13.9 | 4.6 / 4.7 | 53 / 70 | 1.8 / 2.0 | 11.4 / 11.5 | 115 / 116 | 9.5 / 15.4 | 3.0 / 4.7 |
| | Jul | 7 / 7 | 17.7 / 17.4 | 5.0 / 4.9 | 97 / 102 | 4.8 / 5.0 | 11.0 / 10.9 | 119 / 119 | 7.9 / 9.9 | 2.6 / 3.2 |
| | Aug | 20 / 20 | 18.3 / 18.2 | 5.2 / 5.1 | 326 / 327 | 3.9 / 4.0 | 8.9 / 8.9 | 99 / 99 | 8.8 / 17.7 | 3.0 / 6.0 |
| | Oct | 41 / 40 | 11.7 / 11.4 | 6.3 / 6.2 | 540 / 580 | 1.5 / 1.7 | 10.0 / 9.8 | 99 / 97 | 10.5 / 17.9 | 3.1 / 5.2 |
| **NBP** | Jan | 59 / 55 | 5.3 / 5.1 | 7.2 / 7.1 | 484 / 480 | 0.5 / 0.6 | 10.8 / 10.8 | 90 / 90 | – | – |
| | Apr | 28 / 30 | 2.7 / 2.7 | 6.7 / 6.7 | 149 / 158 | 7.8 / 7.8 | 14.9 / 14.9 | 115 / 115 | 7.6 / 9.3 | 1.7 / 2.1 |
| | May | 8 / 8 | 13.8 / 13.9 | 6.8 / 6.7 | 75 / 86 | 1.5 / 1.5 | 11.6 / 11.5 | 119 / 118 | 5.4 / 6.1 | 1.7 / 1.9 |
| | Jul | 11 / 11 | 15.3 / 15.4 | 6.4 / 6.5 | 165 / 179 | 5.0 / 4.6 | 10.5 / 10.4 | 110 / 110 | 5.3 / 6.0 | 1.7 / 1.9 |
| | Aug | 20 / 21 | 18.2 / 18.2 | 6.1 / 6.1 | 243 / 250 | 3.4 / 3.4 | 8.7 / 8.7 | 98 / 98 | 3.9 / 4.5 | 1.3 / 1.5 |
| | Oct | 41 / 39 | 11.8 / 11.7 | 6.9 / 7.0 | 436 / 438 | 1.0 / 1.0 | 9.8 / 9.8 | 98 / 98 | 4.9 / 6.7 | 1.5 / 2.0 |
| **GoR** | Jan | 28 / 34 | 3.6 / 3.4 | 5.8 / 5.8 | 482 / 483 | 1.3 / 1.4 | 11.7 / 12.1 | 93 / 94 | – | – |
| | Apr | 28 / 29 | 3.0 / 3.0 | 5.6 / 5.4 | 180 / 227 | 7.9 / 9.2 | 14.4 / 14.8 | 113 / 115 | 13.7 / 16.6 | 3.1 / 3.7 |
| | May | 5 / 5 | 16.8 / 16.4 | 5.2 / 5.3 | 93 / 148 | 1.7 / 1.7 | 11.4 / 11.3 | 123 / 121 | 6.7 / 12.4 | 2.2 / 4.0 |
| | Jul | 7 / 8 | 17.0 / 17.0 | 5.6 / 5.7 | 204 / 210 | 4.7 / 4.9 | 10.2 / 10.2 | 108 / 110 | 4.5 / 6.1 | 1.4 / 2.0 |
| | Aug | 16 / 15 | 18.9 / 18.9 | 5.4 / 5.6 | 385 / 413 | 4.3 / 4.8 | 8.5 / 8.6 | 97 / 98 | 5.0 / 10.2 | 1.8 / 3.6 |
| | Oct | 32 / 31 | 10.2 / 10.0 | 6.0 / 6.0 | 519 / 495 | 2.6 / 3.5 | 9.7 / 9.7 | 95 / 94 | 5.3 / 7.3 | 1.5 / 2.0 |

420



## 4.4 Estimates of the air-sea CO₂ and CH₄ fluxes

The air-sea $CO_2$ and $CH_4$ fluxes calculated for every research cruise were seasonally averaged so that winter is characterised by the cruise in January, spring – April and May, summer – July and August, and autumn – October (Tables 2 and 3; Fig. 11).

**Table 2.** Seasonal and annual $CO_2$ flux estimates with standard deviations for the Estonian sub-basins and sea area in 2018.

| | **Seasonal median / mean CO₂ fluxes** (g C m⁻² day⁻¹; ± standard deviation) | | | |
|---|---|---|---|---|
| | **GoF** | **NBP** | **GoR** | **Estonian sea area** |
| Winter | 0.06 / 0.07 (±0.04) | 0.02 / 0.02 (±0.02) | 0.08 / 0.07 (±0.05) | 0.04 / 0.05 (±0.04) |
| Spring | -0.30 / -0.36 (±0.23) | -0.14 / -0.13 (±0.09) | -0.22 / -0.21 (±0.13) | -0.21 / -0.26 (±0.20) |
| Summer | -0.11 / -0.11 (±0.06) | -0.16 / -0.17 (±0.08) | -0.04 / -0.02 (±0.13) | -0.10 / -0.10 (±0.11) |
| Autumn | 0.29 / 0.32 (±0.20) | 0.04 / 0.04 (±0.03) | 0.13 / 0.14 (±0.14) | 0.16 / 0.19 (±0.19) |
| Annual mean | -0.02 | -0.06 | -0.005 | -0.03 |

**Table 3.** Seasonal $CH_4$ flux estimates with standard deviations for the Estonian sub-basins and sea area in 2018.

| | **Seasonal median / mean CH₄ fluxes** (mg C m⁻² day⁻¹; ± standard deviation) | | | |
|---|---|---|---|---|
| | **GoF** | **NBP** | **GoR** | **Estonian sea area** |
| Winter | – | – | – | – |
| Spring | 0.20 / 0.24 (±0.17) | 0.03 / 0.05 (±0.06) | 0.12 / 0.26 (±0.31) | 0.14 / 0.20 (±0.22) |
| Summer | 0.14 / 0.43 (±0.99) | 0.05 / 0.06 (±0.07) | 0.04 / 0.18 (±0.46) | 0.07 / 0.25 (±0.71) |
| Autumn | 0.37 / 0.74 (±0.77) | 0.07 / 0.10 (±0.12) | 0.07 / 0.21 (±0.29) | 0.18 / 0.39 (±0.59) |

The $CO_2$ flux estimates (Table 2) show that the Estonian sea area was a source of atmospheric $CO_2$ (positive flux) during winter and autumn and a sink (negative flux) during spring and summer 2018. What stands out is that the standard deviations are of the same order of magnitude as the estimated average fluxes. The observed spatial variability was larger in the GoR and GoF than in the NBP (Figs. 4-9). The annual mean flux (estimated as an average of the four seasonal flux estimates) in the NBP was -0.06 g C m⁻² d⁻¹, in the GoF -0.02 g C m⁻² d⁻¹ and in the GoR -0.005 g C m⁻² d⁻¹.

The $CH_4$ flux estimates (Table 3) show that the Estonian sea area was a source of atmospheric $CH_4$ during spring, summer and autumn. Note that the standard deviations of flux estimates exceeded the resulting average fluxes during summer and autumn. As with $CO_2$ fluxes, the spatial variability of observed $CH_4$ fluxes was larger in the GoR and GoF than in the NBP (Fig. 11).




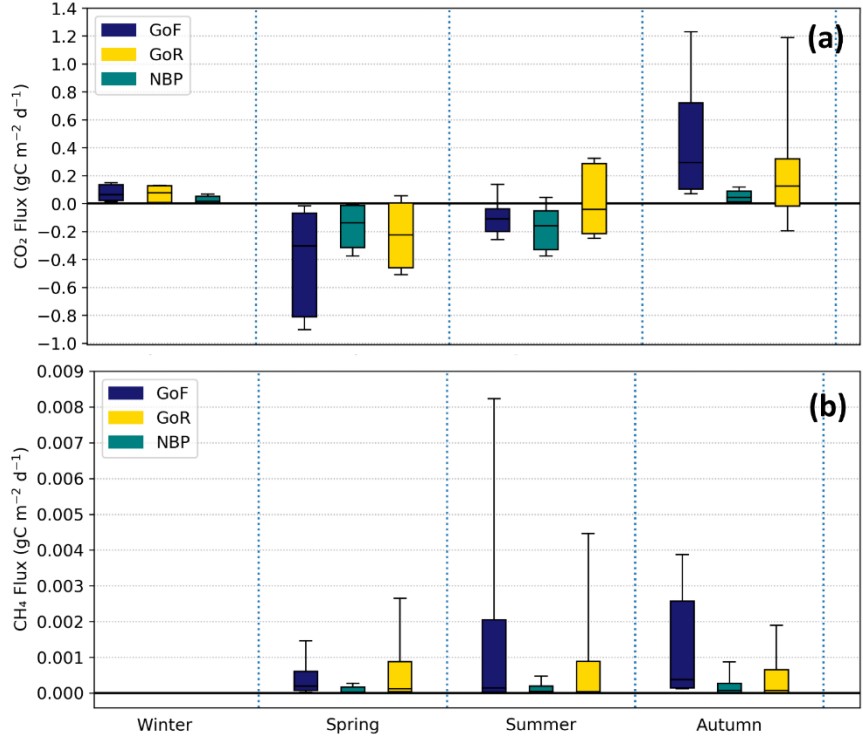

**Figure 11: Seasonal median air-sea (a) $CO_2$ and (b) $CH_4$ flux estimates in the three analysed sub-basins of the north-eastern Baltic**
**Sea in 2018. The flux is positive for the transport from the sea to the atmosphere, while negative values refer to the transport from the atmosphere to the sea.**

## 5 Discussion

The first extensive trace gases study was conducted to describe spatial patterns and seasonal dynamics of $CO_2$ and $CH_4$ in the north-eastern Baltic Sea area, including the GoR. Physical forcing and hydrographic/biological background in the study year

(2018) were analysed to explain the observed patterns in $pCO_2$ and $cCH_4$. The main focus was on the southern Gulf of Finland and the Gulf of Riga, as earlier studies addressed measurements from the Baltic Proper and the western Gulf of Finland (e.g. Schneider et al., 2014; Schneider and Müller, 2018; Gülzow et al., 2013).

### 5.1 Patterns of variability in the southern GoF and GoR

### 5.1.1 $pCO_2$ distribution patterns

Cruises of R/V Salme covered both the offshore and the coastal areas in the GoF and GoR, with the most prominent local $pCO_2$ peaks in the shallow coastal sea areas. These local maxima were mostly linked to coastal physical processes such as





river bulges, coastal upwelling events, fronts and vertical mixing reaching the seabed, but also to phytoplankton distribution patterns influenced by meteorological and hydrographic conditions.

Rivers are the major source of carbon, including dissolved inorganic carbon for the coastal ocean (Dai et al., 2022) and the Baltic Sea (Kuliński and Pempkowiak, 2011). In the GoF, river discharge is mainly concentrated in the easternmost part of the gulf, and the largest freshwater source along the R/V track was the Narva River (Stålnacke et al., 1999) visited in April, May-June and August. In the GoR, river discharge is concentrated in the southern and eastern parts of the gulf (Yurkovskis et al., 1993). Observations close to the largest river discharges in the south of GoR were conducted in May-June, July, August and

October, and in the shallow Pärnu Bay under the influence of the Pärnu River discharge in April, May-June, August and October.

The influence of Narva River waters, identified by a simultaneous local decrease in salinity and increase in $p$CO$_2$, was largest in August (Fig. 8c), when CO$_2$ concentrations locally exceeded saturation level, while in April and May-June, only a slight

increase in $p$CO$_2$ relative to the surrounding waters was observed. The influence of large rivers of the southern GoR was detected by a slight decrease in salinity in May-June and July, but a simultaneous increase in $p$CO$_2$ was observed only in May-June. The largest peaks, which could be linked to the river discharges, were associated with the Pärnu River, especially in April and May-June, when the background levels of $p$CO$_2$ in the adjacent regions were already low. Thus, the river waters influenced the observed $p$CO$_2$ patterns remarkably in the shallow and semi-enclosed Pärnu Bay and less in other areas.

Probably, the research vessel track did not reach the river bulges properly in the southern GoR, or their influence was not seen offshore since the riverine waters were mostly transported along the coast (e.g. Lips et al., 2016b) and an anticyclonic river bulge was not formed as suggested by a modelling study (Soosaar et al., 2016).

Upwelling is the most prominent mesoscale process in the elongated GoF (Myrberg and Andrejev, 2003), where upwelling

events along the southern coast are associated with north-easterly and easterly winds (Lips et al., 2009; Kikas and Lips, 2016). Winds supporting upwelling along the GoF southern coast dominated in October and also, though with lower wind speed, in July and late May. In October, up to three times higher $p$CO$_2$ values than the background were recorded in the upwelled waters. Based on a comparison of salinity and temperature in the surface layer in the upwelling area and vertical profiles registered at the monitoring stations two days earlier, the upwelled waters mostly originated from the water layer of 65-75 m, i.e. the

halocline. Most likely, the observed extreme $p$CO$_2$ values were caused by the relatively deep origin and the impact of the seabed when these waters were brought to the surface along the GoF slope.

In the case of upwelling in the Gotland basin in July-August 2016, a very sharp increase in $p$CO$_2$ was measured, although the absolute values were lower (Jacobs et al., 2021). The authors evaluated the air-sea fluxes of CO$_2$ due to this upwelling event

and showed that the CO$_2$ flux, expected to be directed from air into the sea in August, was reduced and even reversed due to





upwelling. High variability of wind conditions, sea surface temperatures and other factors make it difficult to estimate the total impact of upwelling events on total air-sea fluxes of $CO_2$ precisely. However, Kuss et al. (2006) suggested that roughly 20% of the annual $CO_2$ uptake in the central Arkona Sea could be balanced by $CO_2$ release during occasional upwelling events in the coastal areas, also considering seasonal differences in their impact. Similarly, Norman et al. (2013) estimated that upwelling
events could possibly decrease the Baltic Sea's annual average $CO_2$ uptake by up to 25%. In 2018, the winds favouring upwelling along the northern coast of GoF prevailed. However, we detected the upwelling events in the western GoF along the southern coast during cruises in late May and October. The highest $pCO_2$ values were recorded in upwelled waters in October when autumn mixing contributed to the vertical exchange, and did not trigger production.

The seasonal course in $pCO_2$ in the surface layer is mainly controlled by the primary production in spring/summer and entrainment of waters with high $pCO_2$ due to remineralisation processes during mixed layer deepening in fall (e.g. Schneider and Müller, 2018). The seasonal succession of phytoplankton, however, could be at different stages of development due to varying meteorological and hydrographic background along the 1500 km long measurement track (e.g. Seppälä and Balode, 1999; Lips et al., 2014). A prominent example of such spatial variability was revealed in April when $pCO_2$ and Chl $a$ were
negatively correlated. The highest $pCO_2$ values, corresponding to the saturation level, observed in the parts of GoR and GoF coincided with lower than the background Chl $a$ and oxygen concentrations (Fig. 5d). The lowest $pCO_2$ values were recorded in shallow areas with a warm surface layer with high Chl $a$ concentrations. It is remarkable that in areas of high $pCO_2$, the sea surface temperature was 1-2 °C, meaning below the temperature of the maximum density of approximately 2.5 °C, and the water column was almost fully mixed. The latter was evident from the vertical profiles registered at the monitoring stations
and relatively high surface salinity, especially in the western and central GoR. It can be assumed that when the bloom reached its peak, low $pCO_2$ could also be recorded in this area.

Like the stratification and bloom development in spring, the upper mixed layer deepening and stratification decay could shape the $pCO_2$ distribution patterns in the surface layer in late summer and autumn. By the August cruise, the upper mixed layer
has been deepened to almost 30 m from the values of around 10 m in July. This resulted in high $pCO_2$ values in shallow areas (Fig. 8c), where the mixing has reached the bottom layer. It is interesting that a similar pattern was not evident in October. The areas with higher $pCO_2$ values during summer cruises had $CO_2$ levels in October lower than in the deeper areas, where the mixing reached the near-bottom layer later, leaving less time for equilibration with the atmosphere in these deeper areas.

The described seasonal pattern was observed in the Väinameri Sea (northern GoR) with an average depth of 5-10 m. However, this area is very dynamic and relatively strong gradients of oceanographic variables (salinity, nutrients, etc.) are observed (Suursaar et al., 2001). We suggest the following processes responsible for the observed pattern. In April, $pCO_2$ levels were lower than in the adjacent deeper areas due to a warm surface layer and an earlier start of the spring bloom since the phytoplankton mixing depth in shallow areas is determined by the bottom depth and not vertical stratification (Townsend et





al., 1994). In late May and July, the highest $pCO_2$ values were measured at the saltier side of the salinity front in the Väinameri Sea. We suggest that these maxima were favoured by weaker vertical stratification and, consequently, stronger vertical fluxes at the denser side of the fronts (e.g., Kahru et al., 1984). In August, the higher $pCO_2$ values in the shallow Väinameri Sea were related to the vertical mixing, and in October, when oversaturation was observed almost along the entire R/V track, $pCO_2$ was higher in other, deeper areas where the vertical flux (due to continuing upward mixing of deep, $CO_2$-rich waters) was still at a higher level, while a larger fraction of the $CO_2$ from the near-bottom layer had already evaded from the Väinameri Sea.

Another area where saltier Baltic Proper and fresher GoR waters meet and the front develops, is the Irbe Strait, conveying most of the GoR water exchange with the NBP (Lilover et al., 1998). Locally, the lowest $pCO_2$ was measured in connection to the Irbe front in April, probably due to the development of vertical stratification supporting the spring bloom, while in the GoR, stratification was weak, and the bloom had not started yet. Contrarily, a slight local peak of $pCO_2$ in July could be caused by more intense vertical transport of sub-surface waters at the front.

### 5.1.2 $c$CH$_4$ distribution patterns

$c$CH$_4$ was oversaturated in the surface layer during the whole study period, which is typical for the Baltic Sea (Gülzow et al., 2013), with prominent local peaks in the GoF and GoR. These peaks can be related to the same physical processes as the local maxima in $pCO_2$ spatial distribution. However, two major peculiarities of $c$CH$_4$ distribution can be noticed – the local maximum values were more than an order of magnitude larger than the background $c$CH$_4$ level, and these prominent maxima were confined to the shallow areas. Almost in all shallow bays, which were visited during different cruises in the GoF and GoR, and also in the Tagalaht Bay in the NBP in October, high peaks of $c$CH$_4$ were recorded. This agrees with the earlier results that methane concentrations and variability are high in shallow coastal areas (Roth et al., 2022; Borges et al., 2016).

The seafloor in most of the shallow bays is characterised by clay and mud or mixed sediments (mud and sand; EMODnet Geology), which are potential internal sources of methane (e.g. Humborg et al., 2019). This explains why $c$CH$_4$ peaks were recorded in these shallow areas where the water column was usually mixed until the seabed. In October, when high $c$CH$_4$ values were also measured in the deeper Kolga Bay and along the relatively deep south-western GoF (Fig. 9e), these findings can be related to autumn vertical mixing and an intense upwelling event. A similar impact of upwelling has also been shown by earlier measurements in the Baltic Sea (Gülzow et al., 2013; Jacobs et al., 2021) and other coastal sea areas (e.g. Kock et al., 2008). It is noteworthy that our data show the impact of upwelling on surface $CO_2$ and $CH_4$ concentrations in fall, when upwelling was identified by salinity rather than temperature. Previous studies (Gülzow et al., 2013; Schneider et al., 2014; Jacobs et al., 2021) use the drop in sea surface temperature as an indicator for upwelling and upwelling-induced greenhouse gas fluxes, which bears the risk of underestimating the importance of upwelling for locally enhanced $CO_2$ and $CH_4$ fluxes in the Baltic in fall.





Elevated $c$CH$_4$ was almost always measured in the Väinameri Sea and can be explained by vertical mixing and resuspension of bottom sediments. Resuspension events occur due to wind mixing and waves but also due to frequently appearing strong

currents in the straits (Suursaar et al., 2001). These strong currents are forced by wind-induced differences in the sea level between the adjacent basins (Otsmann et al., 2001). However, the detected $c$CH$_4$ peaks in the Väinameri Sea were not as strong as in the Pärnu Bay, for instance, and in October, a much higher cCH$_4$ peak was measured just outside of the Väinameri in the upwelled waters in the south-western GoF (concentrations reached up to 70 nmol L$^{-1}$; Fig. 9e).

In the Irbe Strait, local $c$CH$_4$ maxima were also frequently observed. However, their locations were different from the observed $p$CO$_2$ extrema. We suggest that also here, the $c$CH$_4$ maxima were related to the shallowest spots along the vessel track (either close to the Kolka or Sõrve peninsulas) and not to the Irbe front, as was suggested for the observed $p$CO$_2$ peaks.

Rivers have been identified as potentially strong sources of CH$_4$ in the Baltic Sea (Myllykangas et al., 2020). Rivers receive

CH$_4$ from soils, groundwater, wetlands, and floodplains in the watershed (De Angelis and Lilley, 1987; Richey et al., 1988). In April and May, elevated $c$CH$_4$ was measured near the Narva River mouth. However, the enhanced $c$CH$_4$ was also measured along the shallow coastal sea towards the west from the Narva River mouth. Similar distribution in August, with the maximum not at the mouth area but in the west, suggests that the river discharge was transported along the coast, as it could occur during summer months (Laanearu and Lips, 2003) or the shallowness and influence of sediments was the main factor creating this

$c$CH$_4$ maximum. The latter suggestion or a combination of both are likely explanations, since the water column was fully mixed along the ship track in this area.

Notable $c$CH$_4$ spatial variability emerged in the shallow Pärnu Bay, while only slightly higher than background $c$CH$_4$ values were measured in the southern GoR, close to the largest river discharges, most probably, as suggested above, due to the absence

of the predicted river bulge (Soosaar et al., 2016). The highest $c$CH$_4$ peaks in the Pärnu Bay were observed in May and August (approximately 200 nmol L$^{-1}$), while in October, the peak was not so prominent, although the river discharge was larger than in summer months. In shallow coastal areas, high methane emissions have been linked to the amount of organic matter in the sediment and water temperature (e.g. Heyer and Berger, 2000). Also, local maxima of $c$CH$_4$ were more pronounced in the Pärnu Bay in comparison with the areas close to the Narva River mouth. We suggest this pattern is related to the shallowness

and semi-enclosed shape of the Pärnu Bay and not directly to the amount and changes in the river runoff.

In summary, physically disturbed organic-rich sediments, river plumes, and upwelling were identified as processes causing hot spots of methane emission. While methane from undisturbed organic-rich sediments usually does not surpass effective anaerobic and aerobic methane oxidation in the upper sediment (e.g. Knittel and Boetius, 2009), physical shear stress can lead

to the release of methane from the upper sediment layer. As a consequence, Borges et al. (2016) suggested water depth as a proxy for methane flux over organic-rich sediments in the North Sea. Loads from rivers are expected to cause the highest





methane inputs in summer because of enhanced organic matter turnover and methanogenesis under warmer conditions, in accordance with our findings. Our data support the importance of shallow water processes for the assessment of the $CH_4$ fluxes from the Baltic (and other marginal seas) to the atmosphere.

## 5.2 Comparison of seasonal variability between the basins

The temporal variations in $pCO_2$ in the surface layer of the NBP, GoF and GoR in 2018 followed, in general, the known seasonal course: high values in winter, low values in spring-summer and a subsequent increase in autumn (Thomas and Schneider, 1999; Schneider and Müller, 2018). The seasonal amplitude of $pCO_2$ was similar in all basins, with a slightly larger range in the GoF. This higher amplitude is likely, at least in part, a result of the lower alkalinities in the GoF, which result in a higher $pCO_2$ change per amount of fixed carbon (e.g. Kulìnski et al., 2017). The seasonal $pCO_2$ minimum in all basins did not coincide with the peak of the spring phytoplankton bloom but appeared at the end of May when surface layer Chl *a* concentrations were already relatively low, as a result of the cumulative nature of the imprint of primary production on the inorganic carbon system. Our data did not reveal an increase in $pCO_2$ between the spring bloom and the cyanobacteria bloom in midsummer, which has been reported based on measurements with a higher temporal resolution (e.g. Schneider et al., 2014).

$pCO_2$ seasonality is mainly controlled by biological activity in combination with the vertical mixing and stratification of the water column (Schneider et al., 2014). In both basins, the GoF and the GoR, Chl *a* concentrations in spring (Fig. 10e, Table 1) were higher than in the NBP, which is in accordance with the elevated nutrient concentrations in the GoF and GoR (HELCOM, 2018). However, a slightly higher average $pCO_2$ in the GoR during spring and summer could be related to the shallowness of this basin. Higher $pCO_2$ in the GoF in October and January can be explained by high biomass production in spring-summer and the specific hydrographic conditions supporting vertical transport and mixing of $CO_2$ from the deep layers in autumn-winter and during the upwelling events, in combination with the reduced buffering due to lower alkalinity in the GoF. High concentrations of inorganic carbon in the deep layers of the GoF result from the organic matter degradation in the presence of stratification in spring and summer and the advection of deep waters from the Baltic Proper (Lehtoranta et al., 2017). In late autumn and winter, collapses of vertical stratification could occur in the GoF (Liblik et al., 2013), resulting in the vertical transport of nutrients (and inorganic carbon) from the near-bottom layer to the surface layer (Lips et al., 2017). The high $pCO_2$ values attributed to upwelling in October in the GoF, also characterized by the lowest surface $O_2$ saturation of the entire survey, confirm active transport mechanisms of deep waters with strong biogeochemical indicators of mineralisation of organic matter (Fig. 9c).

Since the GoR is shallower than the GoF and without a permanent halocline, $CO_2$ accumulation and subsequent $CO_2$ flux from the GoR deep layer do not have a similar high potential in autumn-winter as in the GoF. However, the seasonal thermocline was stronger in spring-summer 2018 than on average due to high heat flux and calm wind conditions (Stoicescu et al., 2022). A near-bottom hypoxic layer developed, and the autumn-winter mixing had not reached the seabed in the deeper central GoR





yet by the cruise in October (Stoicescu et al., 2022, also Fig. 9a), and this could be a reason that relatively low $pCO_2$ was measured in the GoR while upwelling-related high values were recorded in the south-western GoF.

Our $pCO_2$ annual dynamics did not reveal second $pCO_2$ minima during summer, which in the NBP have been reported to appear in early July (Schneider and Müller, 2018). This is most probably due to the long interval between cruises: 5 weeks
between the cruises in late May–early June and mid-July and 5 weeks between the cruises in mid-July and the end of August. Summer cyanobacterial bloom in the GoF is usually starting at the end of June or early July (Lips and Lips, 2008). Based on our data from mid-July, calm wind conditions and high sea surface temperature (median 17.7 °C, Table 1) favoured the bloom, which is also observable in an increase in Chl $a$ concentration. Müller et al. (2021) showed that in 2018, in the Eastern Gotland Sea, the production was intense from the beginning of their study period, the 6th of July (surface water $pCO_2$ was already as
low as around 100 µatm), and *Nodularia* sp peaked on the 24th of July, which is also the date of lowest $pCO_2$ values around 70 µatm. However, in the GoR, cyanobacteria biomass was three times lower in 2018 than in 2017 and a factor of two lower than the long-term mean (Kownacka et al., 2022).

$c$CH$_4$ dynamics in 2018 in the NBP, GoF and GoR follow the general seasonal cycle with the lowest methane concentrations
in summer. $c$CH$_4$ seasonality is controlled by the sediment organic matter content in combination with the stratification of the water column. Gülzow et al. (2013) showed that the GoF surface water is characterised by elevated methane concentrations throughout the year compared to other Baltic Sea regions (their study area did not cover the GoR). During summer, thermal stratification of the water column hampers the methane transport between the sea surface and the deeper layers. The surface water gets depleted in methane due to loss to the atmosphere by sea-air exchange, in part driven by the temperature-induced
decreasing solubility. Schmale et al. (2010) suggested that during summer, elevated methane concentrations are observable in the GoF water column up to a depth of 20–30 m. Aerobic methane production has also been demonstrated to contribute to the slight oversaturation of surface waters in the central Baltic Sea (Schmale et al., 2018; Stawiarski et al., 2019), but the clear link of methane peaks to shallow areas and episodes of mixing reaching the seafloor suggests that these processes are of minor importance in our study area.


In our study, the NBP summer minimum remained within the comparable range with Gülzow et al. (2013), but the GoF summer minimum was twice as high. We also covered the shallow southern coastal sea areas of the GoF with remarkable local peaks, while the GoF sub-transect by Gülzow et al. (2013) was almost fully located in the central Gulf of Finland and avoided shallow coastal waters. Furthermore, the highest CH$_4$ concentrations observed in October 2018 in the south-western GoF could be a
consequence of the specific hydrographic conditions – a combined effect of strong upwelling and autumn mixing.

In the GoR, sediments have a high organic matter content, and the area undergoes intermittent seasonal hypoxia (Stoicescu et al., 2022). In the shallow areas, likely wind-induced mixing remains relevant for the transport of methane from the sediment,





including the potential for sediment resuspension and mobilization of methane-enriched pore waters. In April, the highest
median $c$CH$_4$ was detected in the western and central parts of the GoR, where the water column was fully mixed down to the
seabed. The seasonal stratification in the GoR during spring-summer 2018 was stronger than on average. It restricted vertical
mixing and led to pronounced near-bottom oxygen depletion (Stoicescu et al., 2022) and probably to relatively low $c$CH$_4$ in
the surface layer of the deeper GoR areas (where the seasonal thermocline existed). Relatively low methane concentrations in
October suggest that the autumn mixing had not yet reached the seabed in deep GoR areas by October.

**5.3 Air-sea gas exchange**

Several approaches have been used to assess whether the Baltic Sea is a sink or source of atmospheric $CO_2$, but no uniform
consensus has been reached regarding the results (Dai et al., 2022). The estimation of fluxes on regional or global scales
depends on the applied approaches, among them, whether $p$CO$_2$ data are calculated from other parameters (i.e. pH and total
alkalinity) or direct $p$CO$_2$ measurements are conducted, model-based or remote sensing approaches used (e.g. Wesslander et
al., 2010; Schneider et al., 2014; Kuliński and Pempkowiak, 2011; Parard et al., 2017). In addition, most of the evaluations
have been performed based on the data in the Baltic Proper (Gotland basin; e.g. Thomas and Schneider, 1999; Schneider et al.,
2014), and only a few studies have included data from the north-eastern sea areas (e.g. Honkanen et al., 2020).

The $CO_2$ flux estimates (Table 2) show that the Estonian sea area was a source of atmospheric $CO_2$ during the winter and
autumn and a sink during the spring and summer of 2018. Also, the estimates suggest that all studied sub-basins were $CO_2$
sinks on an annual basis. However, due to the high temporal and spatial variability and the fact that the 6 cruises were
distributed unevenly across the year, with bi-monthly gaps between the measurements in autumn and winter, it cannot be
conclusively defined whether the area is a source or a sink over the course of the year.

The results of our study show no major differences in the behaviour between the three basins. The high variability in the GoR
and GoF was likely observed due to the fact that more shallow coastal areas were covered in these regions. It was also pointed
out by Gutiérrez-Loza et al. (2021) that the fluxes in the Baltic Sea coastal regions were larger than in the open sea area. As
seen in our results, surface water $p$CO$_2$ distribution showed most prominent local $p$CO$_2$ peaks in the shallow coastal sea areas,
which were mostly linked to coastal physical and biogeochemical processes. These mechanisms must also be considered as
modulating the efficiency of the gas transfer across the air-sea interface. Therefore, determining the reasons for the variability
of the surface water $p$CO$_2$ distribution is essential to accurately describe the flux estimates and their role in the carbon budget
of the corresponding region.

In our study, the estimated annual mean flux in the NBP was -0.06 g C m$^{-2}$ d$^{-1}$ (-1.8 mol m$^{-2}$ yr$^{-1}$). Flux estimates for 2005,
2008 and 2009 by Schneider et al. (2014) showed that the central and northern Gotland basins act as a net sink for atmospheric
$CO_2$ with uptake rates ranging between -0.60 and -0.89 mol m$^{-2}$ yr$^{-1}$. Several factors may account for these differences in flux



estimates. The summer of 2018 could have been more productive due to warm weather conditions. On the other hand, our measurements covered the transition areas NBP–GoF and NBP–GoR, where the fluctuations in fluxes are greater than in offshore areas analysed in the study by Schneider et al. (2014). Müller et al. (2021) concluded that their observations in the eastern Gotland basin in July–August 2018 were representative for Baltic Sea cyanobacteria blooms in general, although the $p$CO$_2$ levels in 2018 varied between the upper and lower ends of the conditions observed in previous years (Schneider and Müller, 2018). Additionally, the difference in flux estimates might be caused by different parametrizations used. Wesslander et al. (2011) showed that the flux estimates in the coastal region off the east coast of Gotland differed by 64% depending on the parametrization used. The differences in the estimated CO$_2$ fluxes using different parametrizations could be largest during the winter months (October to February), as suggested by Gutiérrez-Loza et al. (2021).

The calculated annual mean fluxes in the GoF and GoR were smaller than in the NBP. An analysis of these values in more detail (Table 2) reveals that CO$_2$ uptake during spring-summer was the greatest in the GoF (Fig. 11a). As a counterbalance to summer absorption, the CO$_2$ release in October had a large impact on the estimates of the annual mean fluxes. In the NBP, the impact of the autumn release was smaller (Table 2), but it was significant for the GoF and GoR annual mean flux estimates due to the upwelling event along the southern coast of the GoF and shallower basin with mixing reaching the seabed in most of the GoR.

The Baltic Sea is a source of atmospheric CH$_4$ and shows strong spatial and seasonal variations (Bange et al., 1994; Gülzow et al., 2013). Also, the present dataset shows that the Estonian sea area is a source of atmospheric CH$_4$ during spring, summer and autumn (Table 3; Fig. 11b). A considerable increase in the calculated methane flux was observed in August, as also pointed out in the study by Gülzow et al. (2013), whereas their study covered mostly the Baltic Sea offshore areas and the increase was explained as a consequence of the transition to the regime of high wind velocities. Due to the upwelling event in October, methane outgassing in our study was most probably intensified in autumn (Jacobs et al., 2021). Our calculated CH$_4$ fluxes in the NBP (Table 3; Fig. 11b) were much lower and much less variable than in the GoF and GoR. The reason is similar, as discussed regarding the CO$_2$ fluxes – more shallow coastal areas were covered in the GoF and GoR than in the NBP.

For a robust flux estimate in the entire (north-eastern) Baltic Sea, understanding and monitoring of coastal processes seem to be mandatory in addition to measurements in the central Baltic – even when integrating over the surface area. On the other hand, even few data from the NBP are likely representative of a very large area, given the error ranges.

## 6 Conclusions

Spatial patterns and seasonal dynamics of CO$_2$ and CH$_4$ were studied in the north-eastern Baltic Sea area. We observed that the southern GoF and GoR have considerably higher spatial variability and seasonal amplitude of surface layer $p$CO$_2$ and $c$CH$_4$



than measured in the Baltic Sea offshore areas. The main processes behind this high variability are coastal upwelling events,
hydrographic fronts (e.g. Irbe front), mixing reaching the seabed and possible shifts in the timing of bloom events influenced
by hydrography. On average, the $CO_2$ air-sea fluxes in the north-eastern Baltic Sea are similar between the sub-basins but with
larger amplitudes in the coastal areas. However, regional variations in $CO_2$ dynamics also result in differences in annual flux
estimates between the sub-basins.

Due to the observed high variability, it is recommended to continue similar high-resolution measurements in the coastal and
offshore areas at least every season during the regular environmental monitoring cruises. It is essential for accurately evaluating
the role of this region in the Baltic Sea carbon budget and to predict potential future changes due to anthropogenic/climatic
pressures. Additionally, high-resolution $p$CO$_2$ measurements have a strong potential to contribute to eutrophication monitoring,
enabling quantitative assessment of organic matter production and mineralisation (Schneider and Müller, 2018), and can be
used as a pivotal parameter to trace acidification.

*Data availability.* The data set will be made available in an openly accessible database upon acceptance.

*Author contribution.* SL, EJ, GR and UL conceived the study. UL contributed to developing methods and writing the
manuscript. GR contributed to developing methods and reviewing the manuscript. EJ contributed by analysing the data and
writing and reviewing the manuscript. STS contributed by analysing and visualizing the data and reviewing the manuscript.
SL carried out analyses, prepared the figures, and wrote the manuscript with editorial and scientific contributions from all the
co-authors.

*Competing interests.* The authors declare that they have no conflict of interest.

*Acknowledgements.* The authors would like to thank Michael Glockzin for his contribution and support with the trace gas
measurements, Villu Kikas for his technical assistance, and the crew of the R/V Salme for their help and cooperation. Silvie
Lainela´s sincere words of thank you also go to TalTech colleagues Sander Rikka and Ilja Maljutenko for the fruitful
discussions in developing and optimising Python and MATLAB scripts and Germo Väli for his help and advice regarding
meteorological data. The study was in the frames of the BONUS INTEGRAL project (grant no. 03F0773A), which received
funding from BONUS (Art 185), funded jointly by the EU, the German Federal Ministry of Education and Research, the



Swedish Research Council Formas, the Academy of Finland, the Polish National Centre for Research and Development, and the Estonian Research Council. The work of Silvie Lainela, Stella-Theresa Stoicescu and Urmas Lips was supported by the Estonian Research Council grant (PRG602). The analysis was also partly supported by the JERICO-S3 project (funded by the European Commission's Horizon 2020 Research and Innovation programme under grant agreements No 871153 and 951799).

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
