# Peer review of "Seasonal dynamics and regional distribution patterns of CO2 and CH4 in the north-eastern Baltic Sea"

_EGUsphere, 2024_

## Author Response (AR1)

**Author´s Response to Reviewer #1**

Description of changes made in the manuscript is denoted with blue.

**GENERAL COMMENTS**

**The paper extensively describes a detailed data-set of CO2 and CH4 in surface waters of NE Baltic Sea. The figures are of good quality but the text could be improved. The terminology is in place awkward. The first part of the Introduction is a succession of unrelated statements.**

**MAJOR COMMENTS**

**Most of the paper is based on textual descriptions of changes of CO2/CH4 that are stated to relate to changes of salinity, temperature, or depth. It could useful to plot CO2/CH4 as a function of these variables to back these statements. Such plots allow to explore possible additional features in the data-set.**

Thank you very much for the comments. If the total data set is considered, simple property-property plots do not show up here. Therefore, we will select certain episodes (salinity for rivers, temperature for upwelling, water depth for sedimentary interaction) and provide some property-property plots in the supplementary material to support and illustrate our statements.

If the total data set is considered, simple property-property plots do not show up here (Figs. 1-5). For instance, high $c$CH$_4$ are more often observed in the shallowest areas, but high values also occur in deep areas under certain conditions (Fig. 1; relevant conditions are explained in the text of the manuscript). Another idea was to show certain episodes (salinity for rivers, temperature for upwelling, water depth for sedimentary interaction) in the supplementary material to support and illustrate our statements (e.g. upwelling influence shown in Figs. 2-5). Unfortunately, the request by the Reviewer #2 to shorten the overall manuscript conflicted with adding these figures to the manuscript or supplemental information.

[Figure]

**Figure 1: $c$CH$_4$ vs depth in the shallow areas during all cruises in 2018.**

[Figure]

**Figure 2: $p$CO$_2$ vs temperature during all cruises in 2018.**

[Figure]

**Figure 3: $p$CO$_2$ vs salinity during all cruises in 2018.**

[Figure]

**Figure 4:** *c*CH₄ vs temperature during all cruises in 2018.

[Figure]

**Figure 5:** *c*CH₄ vs salinity during all cruises in 2018.

**It's unclear why the mixed layer depth is used to explain patterns in CO2 and CH4. I suggest that the authors compute a stratification index such potential energy anomaly (PEA) according to Simpson (1981). This is a simple computation from the density (salinity-temp) vertical profiles that allows to quantify the strength of water column stratification.**

Thank you for the suggestion. However, we are of the opinion that calculating PEA will not be fully relevant in here, as it will depend on the depth. The idea here was to show whether the mixing reaches the bottom or not, stimulating enhanced bottom shear stress and thus sediment interaction. We will add a short explanation to give clearer reasoning for the choice of mixed-layer depth in this context.

In the section 3.4 CTD profiles and upper mixed layer depth, we changed the sentence accordingly: "The depth of the UML was determined from the CTD profiles at the monitoring stations to evaluate whether vertical mixing reached all the way down to the seabed. It was done by comparing the UML depth with the water depth along the ship track adjacent to each station.".

**SPECIFIC COMMENTS**

**I do not see what is the logical link between the first, second and third paragraphs of the introduction. The content is correct, but it's unclear how these statements connect together to introduce the paper. I suggest to remove the first two paragraphs and start the introduction by the section on the Baltic.**

With the Introduction's first paragraphs, our point was to provide the reader with a general context of carbon system parameters in the atmosphere, then on the air-sea interface and finally describe the processes in the seawater in more detail. To improve the coherence of the Introduction´s text, after the first paragraph we will add a paragraph of the role of the marine realm and more specifically the coastal ocean in the $CO_2$ / $CH_4$ cycle and atmospheric budget and improve the logical connection along this line.

After the Introduction´s first paragraph we added an additional paragraph: "The global ocean is estimated to be a net sink of $CO_2$ (26 % of total $CO_2$ emissions during the decade 2012-2021; Friedlingstein et al., 2022). However, these global estimates are only beginning to resolve the net $CO_2$ source/sink characteristics of the coastal ocean. The complexity of processes in the coastal ocean and the limited data availability make it difficult to quantify regional carbon budgets and the coastal ocean's role in the global carbon budget. Although oceanic methane emissions play a modest role in the global methane budget (Reeburgh, 2007), estuaries and other coastal areas contribute up to 75% of all oceanic $CH_4$ emissions (Bange et al., 1994), with an important but not well quantified contribution of very shallow waters (Borges et al., 2016).".

**L44: There are recent papers showing long term changes in salinity and alkalinity that should also affect the "CO2 system of surface waters in the Baltic Sea".**

Thank you. We will modify the sentence: "In addition to the exchange at the air-sea interface and biological processes, the $CO_2$ system of surface waters in the Baltic Sea is influenced by the changes in hydrological and hydrographic conditions, e.g. river discharges, waves, currents, salinity and

temperature, vertical stratification and mixing, upwelling/downwelling, fronts, etc. (e.g. Müller et al., 2016; Jacobs et al., 2021).".

We changed the sentence accordingly: "In addition to the exchange at the air-sea interface and biological processes, the $CO_2$ system of surface waters in the Baltic Sea is influenced by the changes in hydrological and hydrographic conditions, e.g. river discharges, waves, currents, salinity and temperature, vertical stratification and mixing, upwelling/downwelling, fronts, etc. (e.g. Müller et al., 2016; Jacobs et al., 2021).".

**L55: The collapse of phytoplankton blooms and delivery of fresh material to sediments poor in organic matter seem to stimulate CH4 release and a seasonal peak that does not coincide with the peak in temperature (Borges et al. 2018). Temperature seems to control seasonality in sediments rich in organic matter.**

We slightly extended the sentence to: "In coastal areas, dominant controlling factors for the seasonal variations of methane emission are the sediment organic matter content (Heyer and Berger, 2000), which might be modulated by seasonal deposition of fresh organic material from primary production, and temperature (Borges et al., 2018).". This is meant as a general statement in the context of $CH_4$ dynamics in shallow and deeper areas. We prefer not to specify the seasonal course and dependencies further in this part (Introduction).

We changed the sentence accordingly: "In coastal areas, the dominant controlling factors for the seasonal variations of methane emission are the sediment organic matter content (Heyer and Berger, 2000), which might be modulated by seasonal deposition of fresh organic material from primary production, and temperature (Borges et al., 2018).".

**L60: Production of methane in aerobic conditions seem to be only relevant in the deep ocean but not the coastal ocean (Weber et al. 2019), although, concentrations and emissions of CH4 in the deep ocean are negligible compared to the coastal ocean.**

This is a little difficult in the context of the Baltic Sea as the mentioned studies are from the central Gotland Basin, and though showing that the process of production by zooplankton cannot explain the observed (moderate) surface oversaturation. However, the modelling analysis suggests that other surface production pathways must play a role here. However, is correct that the contribution from aerobic production is not important in methane-rich coastal areas. We modify our sentence

accordingly: "Production in the upper, oxygenated water column might also contribute to or even govern methane sea-air fluxes (Schmale et al., 2018; Stawiarski et al., 2019), but is of minor importance in the coastal ocean (Weber et al., 2019), and negligible in shallow coastal areas of high methane concentrations / emissions.".

We changed the sentence accordingly: "Production in the upper, oxygenated water column might also contribute to or even govern methane sea-air fluxes (Schmale et al., 2018; Stawiarski et al., 2019), but it is of minor importance in the coastal ocean (Weber et al., 2019) and negligible in shallow coastal areas of high methane concentrations / emissions.".

**I suggest that the authors use the full names of the regions instead of the abbreviations (GoF, GoR, and NBP). In the journal Biogeosciences, there is no word limit, so it is unnecessary to abbreviate. For the readers that are unfamiliar with the Baltic Sea it is already difficult to follow the reasoning with these different sub-regions. The use of abbreviations leads to further confusion (letter soup).** Thank you for this suggestion, but we do not consider it necessary to use full names instead of abbreviations since there are only three abbreviations (GoF, GoR and NBP) which are used throughout the manuscript and plots. Using the full names of the regions makes the overview of already detailed plots even more confusing.

We kept the three abbreviations and use them throughout the manuscript and plots.

**L200: why "rapid" ? in relation to what ?** We believe that this comment refers to line 209. 'Rapid model' is the name of the approach used for the gas flux calculations (described in detail by Woolf et al., 2016). The term is also used in the FluxEngine vocabulary (Holding et al., 2019). We will modify the sentence: "The $CO_2$ fluxes were calculated using a rapid model approach (Woolf et al., 2016) implemented into the FluxEngine toolbox.".

We changed the sentence accordingly: "The $CO_2$ fluxes were calculated using a rapid model approach (Woolf et al., 2016) implemented into the FluxEngine toolbox.".

**L210: It is not necessary to use subscripts A and W for Henry's constant ($\alpha$). The same value is applied to both pCO2 in air and water.**

The 'rapid' equation takes into account solubilities in the skin layer and below the surface layer (in the foundation layer); therefore, different subscripts are used.

In the manuscript, after the equation (1), we will complement the sentence and explain the meaning of the subscripts more precisely.

We changed the description of equation (1) accordingly: " where $F$ (g C m$^{-2}$ day$^{-1}$) denotes the flux across the interface, $k$ the gas transfer velocity, $\alpha$ the solubility of gas in the subsurface water and the water surface (subscripts $W$ and $A$, accordingly) and $p$CO$_2$ partial pressure of CO$_2$ in the sea surface water/atmosphere (subscripts $W$ and $A$, accordingly).".

**L216: Equation (2) is incorrect. cCH4 corresponds to dissolved concentration so Henry's constant (α) is not necessary and in fact chemically meaningless.**

Thank you for this comment. We will correct the equation (2).

Equation (2) was corrected accordingly: $F = k(cCH_{4_W} - cCH_{4_A})$.

**L222: This is a strange result. Please briefly explain why "negligible differences in the average net CO2 flux were observed when using the different gas transfer parametrisations". I guess this reflects that wind speed was generally low since all parameterisations converge at low wind speed. Please list the different gas transfer parametrisations that were tested.**

We will clarify the paragraph accordingly: "In order to accurately describe the fluxes and the carbon budget, it is essential to include relevant processes to the air–sea CO$_2$ and CH$_4$ flux parametrisation. Nightingale et al. (2000) was used for the gas transfer velocity parametrisation for both CO$_2$ and CH$_4$ in our study. The sensitivity analysis of the gas transfer velocity in the Baltic Sea (Gutiérrez-Loza et al., 2021) used different parametrisations of the gas transfer velocity to evaluate the effect of other relevant processes in addition to wind speed on the net CO$_2$ flux at regional and sub-regional scale. In the Estonian sea area, they observed negligible differences in the average net CO$_2$ flux when using the different gas transfer parametrisations relative to the wind-based parametrisation.".

We changed the paragraph accordingly: "In order to accurately describe the fluxes and the carbon budget, it is essential to include relevant processes to the air–sea CO$_2$ and CH$_4$ flux parametrisation. Nightingale et al. (2000) was used for the gas transfer velocity parametrisation for both CO$_2$ and CH$_4$ in our study. The sensitivity analysis of the gas transfer velocity in the Baltic Sea (Gutiérrez-Loza et al.,

2021) used different parametrisations of the gas transfer velocity to evaluate the effect of other relevant processes in addition to wind speed on the net $CO_2$ flux at regional and sub-regional scale. In the Estonian sea area, they observed negligible differences in the average net $CO_2$ flux when using the different gas transfer parametrisations relative to the wind-based parametrisation: ".

**L 320: it's water that is under-saturated not CO2 itself.**

We will correct our sentence accordingly: "The surface waters were undersaturated in $CO_2$ in most areas of the GoF except the Narva Bay, where the water column was well-mixed down to the seabed, oversaturated in the Väinameri Sea and Pärnu Bay, and undersaturated in the NBP (Fig. 8c).".

We changed the sentence accordingly: "The surface waters were undersaturated in $CO_2$ in most areas of the GoF except the Narva Bay, where the water column was well-mixed down to the seabed, oversaturated in the Väinameri Sea and Pärnu Bay, and undersaturated in the NBP (Fig. 8c).".

**REFERENCES**

Müller, J. D., Schneider, B., and Rehder, G.: Long-term alkalinity trends in the Baltic Sea and their implications for CO2-induced acidification, Limnol. Oceanogr., 61, 1984–2002, https://doi.org/10.1002/lno.10349, 2016.

Added to the list of references.

Weber, T., Wiseman, N. A., and Kock, A.: Global ocean methane emissions dominated by shallow coastal waters, Nat. Commun., 10, 4584, https://doi.org/10.1038/s41467-019-12541-7, 2019.

Added to the list of references.

**Author´s Response to Reviewer #2**

Description of changes made in the manuscript is denoted with blue.

**This is a rather lengthy manuscript for a study in a small area lasting only one year. There is little new information (besides the site-specific data), and the manuscript should be significantly shortened.**

Thank you for your comment, though we naturally do not agree that there is little new information provided in our manuscript. However, we see the point and will shorten the manuscript in the results and discussion sections in the final version.

We tried to shorten the manuscript as much as possible. In total (from the beginning of the Abstract to the end of the Conclusions), 12160 words are in the revised version versus to 12681 in the initial version. The decrease is not very significant since we added quantitative information in several places as requested by the Reviewers.

**The abstract is way too descriptive and has little useful information for readers. Quantitative ones in the abstract and the conclusions should replace qualitative statements.**

Thank you for this comment. We will add quantitative values in the abstract and conclusion sections. However, since the annual estimates are near zero, we prefer to keep the last sentence without a quantitative value.

**Abstract.** Significant research has been carried out in the last decade to describe the $CO_2$ system dynamics in the Baltic Sea. However, there is a lack of knowledge in this field in the NE Baltic Sea, which is the main focus of the present study. We analysed the physical forcing and hydrographic background in the study year (2018) and tried to elucidate the observed patterns of surface water $CO_2$ partial pressure ($pCO_2$) and methane concentrations ($cCH_4$). Surface water $pCO_2$ and $cCH_4$ were calculated from continuous measurements during six monitoring cruises onboard R/V Salme, covering the Northern Baltic Proper (NBP), the Gulf of Finland (GoF) and the Gulf of Riga (GoR) and all seasons in 2018. The general seasonal $pCO_2$ pattern showed oversaturation in autumn-winter (average relative $CO_2$ saturation 1.2) and undersaturation in spring-summer (average relative $CO_2$ saturation 0.5), but it locally reached the saturation level during the cruises in April, May and August in the GoR and in August in the GoF. $cCH_4$ was oversaturated during the entire study period, and the seasonal course was not well exposed on the background of high variability. Surface water $pCO_2$ and $cCH_4$ distributions showed larger spatial variability in the GoR and GoF than in the NBP for all six cruises. We linked the observed local maxima to river bulges, coastal upwelling events, fronts, and occasions when vertical mixing

reached the seabed in shallow areas. Seasonal averaging over the $CO_2$ flux based on our data suggest a weak sink for atmospheric $CO_2$ for all basins, but high variability and the long periods between cruises (temporal gaps in observation) preclude a clear statement.

**6 Conclusions**

Spatial patterns and seasonal dynamics of $CO_2$ and $CH_4$ were studied in the north-eastern Baltic Sea area. We observed that the southern GoF and GoR have considerably higher spatial variability and seasonal amplitude of surface layer $pCO_2$ and $cCH_4$ than measured in the Baltic Sea offshore areas ($pCO_2$ 50-1200 µatm vs 100-550 µatm, respectively; $cCH_4$ 80 vs 22 nmol $L^{-1}$, respectively). The main processes behind this high variability are coastal upwelling events, hydrographic fronts (e.g. Irbe front), mixing reaching the seabed and possible shifts in the timing of bloom events influenced by hydrography. On average, the $CO_2$ air-sea fluxes in the north-eastern Baltic Sea are similar between the sub-basins but with larger amplitudes in the coastal areas. However, regional variations in $CO_2$ dynamics also result in differences in annual flux estimates between the sub-basins.

Due to the observed high variability, it is recommended to continue similar high-resolution measurements in the coastal and offshore areas at least every season during the regular environmental monitoring cruises. It is essential for accurately evaluating the role of this region in the Baltic Sea carbon budget and to predict potential future changes due to anthropogenic/climatic pressures. Additionally, high-resolution $pCO_2$ measurements have a strong potential to contribute to eutrophication monitoring, enabling quantitative assessment of organic matter production and mineralisation (Schneider and Müller, 2018), and can be used as a pivotal parameter to trace acidification.

We added quantitative values in the abstract and changed the sentence accordingly: "The general seasonal $pCO_2$ pattern showed oversaturation in autumn-winter (average relative $CO_2$ saturation 1.2) and undersaturation in spring-summer (average relative $CO_2$ saturation 0.5), but it locally reached the saturation level during the cruises in April, May and August in the GoR and in August in the GoF.".

We added quantitative values in the conclusion section and changed the sentence accordingly: "We observed that the southern GoF and GoR have considerably higher spatial variability and seasonal amplitude of surface layer $pCO_2$ and $cCH_4$ than measured in the Baltic Sea offshore areas ($pCO_2$ 50-1200 µatm vs 100-550 µatm, respectively; $cCH_4$ 80 vs 22 nmol $L^{-1}$, respectively).".

**Minor points:**

**1. It may be confusing to state that pCO2 and cCH4 were calculated from continuous measurements. They were indeed measured, albeit with an intermediate step of converting xCO2 and xCH4 to pCO2 and cCH4.**

Thank you for the comment. To make the sentence in the abstract more clearly understood, we will modify it accordingly: Surface water $p$CO$_2$ and $c$CH$_4$ were continuously measured during six monitoring cruises onboard R/V Salme, covering the Northern Baltic Proper (NBP), the Gulf of Finland (GoF) and the Gulf of Riga (GoR) and all seasons in 2018.

We changed the sentence in the abstract accordingly: "Surface water $p$CO$_2$ and $c$CH$_4$ were continuously measured during six monitoring cruises onboard R/V Salme, covering the Northern Baltic Proper (NBP), the Gulf of Finland (GoF) and the Gulf of Riga (GoR) and all seasons in 2018.".

**2. The pCO2 vertical scale should be modified to eliminate the empty space below 350 in Fig. 4 so that the signals are enlarged.**

Thank you for the comment. The scales between Figs. 4-9 are identical (unless stated otherwise in the figure caption) and were compiled so that the six cruises of the year could be compared more easily. Therefore, we prefer not to modify the scale in Fig. 4, acknowledging that this leads to reduced resolution in some cases.

We did not modify the vertical scale in Fig. 4.